# Archived natural DNA samplers reveal four decades of biodiversity change across the tree of life

Isabelle Junk [1,7], Julian Hans [1,7], Benoît Perez-Lamarque [2,3,7], Manuel Stothut [1], Sven Weber [4], Elisabeth Gold[1], Caroline Schubert[1], Alice Schumacher[1], Nina Schmitt[1,5], Anja Melcher [1], Martin Paulus[1], Roland Klein [1], Diana Teubner[1], Jan Koschorreck[6], Susan Kennedy[1], Hélène Morlon [2] & Henrik Krehenwinkel [1] ✉

Detecting the imprints of global environmental change on biological communities is a paramount task for ecological research. But a lack of standardized long-term biomonitoring data prevents a deeper understanding of biodiversity change in the Anthropocene. Novel sources of data for analysing biodiversity change across time and space are urgently needed. By metabarcoding highly standardized biota samples from a long-term pollution monitoring archive in Germany, we here analyse four decades of community diversity for tens of thousands of species across the tree of life. The archived samples—tree leaves, marine macroalgae, and marine and limnic mussels—represent natural community DNA samplers, preserving a taxonomically diverse imprint of their associated biodiversity at the time of collection. We find no evidence for universal diversity declines at the local scale. Instead, a gradual compositional turnover emerges as a universal pattern of temporal biodiversity change in Germany's terrestrial and aquatic ecosystems. This turnover results in biotic homogenization in most terrestrial and marine communities. Limnic communities, in contrast, rather differentiate across space, probably due to the immigration of different invasive species into different sites. Our study highlights the immense promise of alternative sample sources to provide standardized time series data of biodiversity change in the Anthropocene.

Global ecosystems have experienced unprecedented human-induced change in the past decades[1–3], with serious consequences for ecosystem functioning and resilience, as well as human well-being[4]. Understanding patterns of biodiversity change in the Anthropocene is crucial for its future protection. To achieve this, standardized, long-term and taxonomically broad biomonitoring data are essential[5,6]. Such data, however, are lacking for most taxa and ecosystems[5,7]. Available time series are often short or incomplete and limited to a few target taxa and study locations[7–9].

A promising solution to this problem is offered by environmental specimen banks (ESBs)—long-term pollution monitoring archives that collect indicator organisms from various terrestrial and aquatic ecosystems[10]. The samples are collected according to highly standardized protocols and stored at low temperatures, which ensures that not only the pollutants, but also the sample-associated nucleic acids, are excellently preserved[10,11]. The indicator species collected by ESBs are associated with diverse communities of interacting taxa,

**a**

**b**

*Fagus sylvatica* (1987–2022)
*Populus nigra* (1991–2020)
*Picea abies* (1985–2022)
*Dreissena polymorpha* (1994–2020)
*Fucus vesiculosus* (1985–2021)
*Mytilus edulis* (1985–2020)

- Bacteria
- Algae/protozoa
- Metazoa
- Fungi

**Fig. 1 | Overview of sampling sites and recovered biodiversity from ESB samples from terrestrial, limnic and marine ecosystems spanning a time window from 1985 to 2022. a**, ESB sampling sites of tree leaves, zebra mussels, bladderwrack and blue mussels across Germany. **b**, Total OTU and order level richness (log scale) of different taxonomic groups across the tree of life associated with tree leaves, zebra mussels, bladderwrack and blue mussels. Taxonomic groups are represented by different colours. Icons refer to sampled species.

each of which leaves a trace of its DNA in the sample[12–14]. Recent work has shown that many ESB indicator species are excellently suited as 'natural DNA samplers'[15] for studying their surrounding biota via DNA metabarcoding[12–14]. The long-term archives of ESBs can thus provide the standardized time series data so urgently needed to understand biodiversity change[10,11].

Here we use metabarcoding of samples from the German Environmental Specimen Bank, one of the largest, most technologically advanced and longest-operating ESBs, to reconstruct biodiversity change across broad taxonomic, spatial and temporal scales. Using archived leaves from tree canopies, we characterize communities of canopy-associated fungi, bacteria and arthropods. Samples of a dominant European marine macroalgal species reveal coastal bacterial and animal communities associated with the alga. Finally, marine and limnic mussels provide an imprint of the surrounding bacterial and eukaryotic communities in coastal waters and rivers.

Using these data, we explore common patterns of biodiversity change across the tree of life in terrestrial, marine and limnic habitats in Germany over the past four decades. Recent work has highlighted different responses of biota to changing environmental conditions in the Anthropocene[13,16–23]. We explore the generality of these responses by testing three hypotheses for the temporal variation of biodiversity: (1) stressful conditions may have led to extinctions of species at individual sites, that is, local declines of α-diversity[16,20,23]; (2) alternatively, we test whether losses of species are countered by the immigration of novel taxa, leading to a pattern of biotic turnover (β-diversity)[17,19,22]. This turnover could occur (2a) gradually with the changing environment[17] or (2b) rapidly, when the community reaches a tipping point[19,22]. (3) Finally, we test whether the immigration of species across broad geographic scales, for example, of widespread invasive taxa, leads to a pattern of biotic homogenization, that is, a decrease of β-diversity across space[13,18]. To test these hypotheses, we developed a dynamic model for community assembly based on the equilibrium theory of island biogeography (ETIB)[24], which generates null expectations of diversity trends in the absence of disturbance. We further analyse the inferred patterns within the conceptual framework of Blowes et al.[21].

## Results

### Natural DNA samplers recover biodiversity across ecosystems

We metabarcoded 550 samples of archived natural sampler organisms from three marine, nine limnic and nine terrestrial sites in Germany (Fig. 1a and Supplementary Data 1). Using time series of leaves from trees (1985–2022), marine macroalgae (1985–2021) and marine (1985–2020) as well as limnic mussels (1994–2020), we reconstructed communities associated with these organisms at the time of sampling. Rarefaction and bootstrapping analyses indicated sufficient sequencing depth and sampling size for biodiversity estimations at both local and regional scales (Extended Data Fig. 1 and Extended Data Table 1). Our analysis recovered highly diverse prokaryote and eukaryote communities (Fig. 1b), a total of 66,184 zero-radius operational taxonomic units (OTUs) in 751 orders and 102 phyla. Tree leaves recovered 5,183 OTUs of bacteria in 94 orders, 6,250 fungal OTUs in 113 orders and 3,275 metazoan (mainly arthropod) OTUs in 24 orders. We found 5,474 bacterial OTUs in 101 orders and 787 metazoan OTUs in 78 orders in marine macroalgae. We found 21,266 OTUs of bacteria in 180 orders and 3,551 OTUs of microeukaryotes (mainly algae and protozoa) in 160 orders in marine blue mussels. In limnic zebra mussels, we found 14,292 bacterial OTUs in 184 orders, 5,587 microeukaryote (mainly algae and protozoa) OTUs in 173 orders and 523 metazoan OTUs in 71 orders (Fig. 1b and Supplementary Data 2).

The detected communities accurately represented their respective ecosystems and natural sampler organisms (Fig. 2 and Supplementary Data 3). For example, various typical coastal metazoans were found in bladderwrack samples and numerous OTUs of eukaryotic algae reflect the phytoplankton community surrounding mussels. Typical canopy-dwelling arthropods and leaf-associated fungi and bacteria were recovered from leaves (Supplementary Data 3). The OTUs detected in tree canopies showed a high specificity for their respective host tree, with each tree species harbouring a unique community of leaf-associated taxa (Extended Data Fig. 2). Typical monophagous arthropods or host-specific microorganisms were exclusively found on their respective host tree (Supplementary Data 3). The communities recovered from all sample types were also site-specific, with many taxa limited to single sites, probably indicating their specific

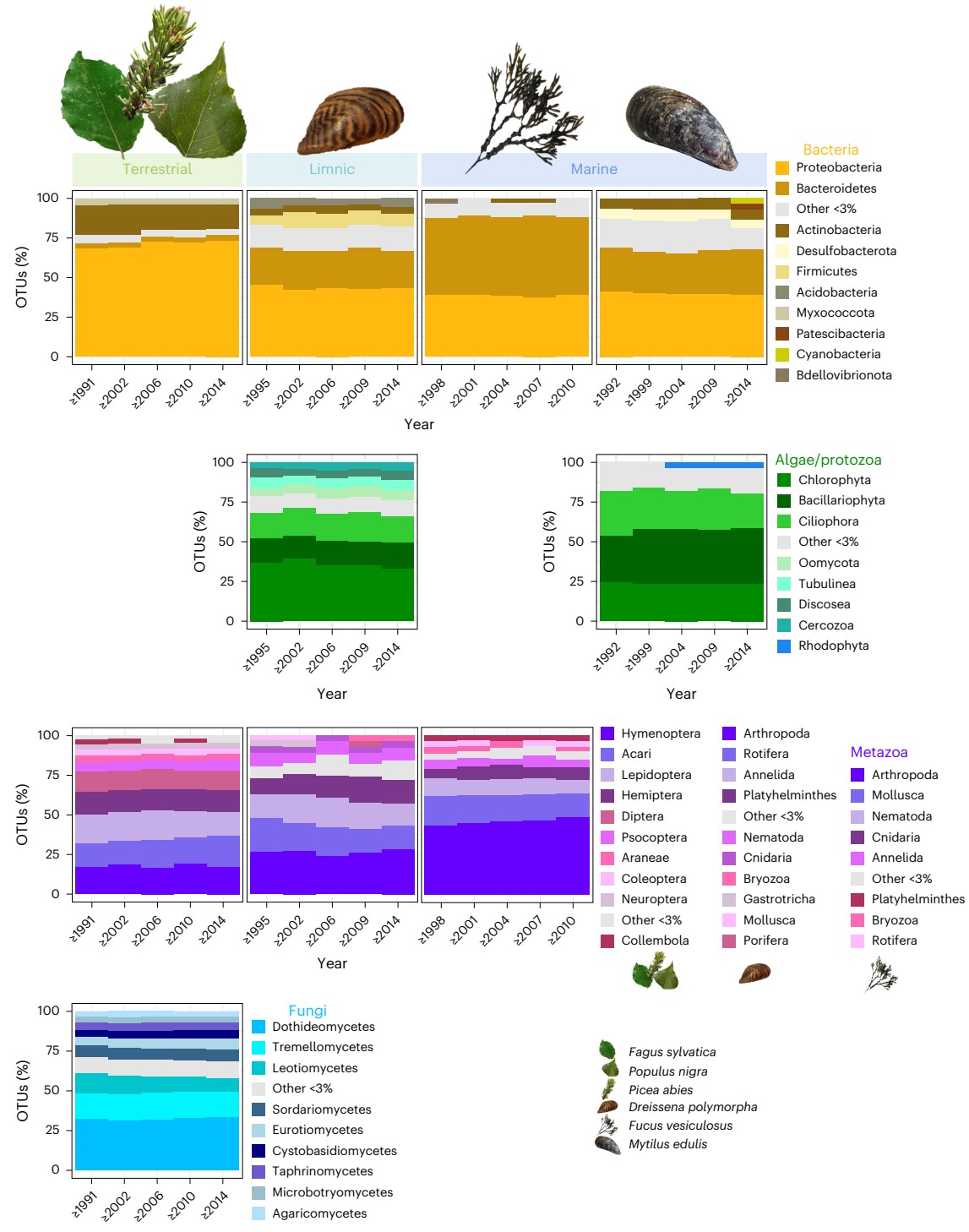

**Fig. 2 | Taxonomic composition of bacteria, microeukaryotic plankton, metazoa and fungi detected in ESB samples from 1991 to 2021.** Taxonomic composition is based on OTU occurrence and is shown for five time periods, each spanning several consecutive years per sampling type. Community composition is shown at phylum level for bacteria, microeukaryotic plankton and aquatic metazoans. Terrestrial metazoans are shown at order level and fungi at class level. Taxonomic groups are represented by different colour palettes. Icons refer to sampled species.

ecological requirements or biogeographic affinities. For example, highly disparate communities were associated with bladderwrack and blue mussels from the Baltic versus the Northern Sea; these two seas are distinguished by pronounced salinity differences[25]. Additionally, our two Northern Sea sampling sites, which are separated by about 200 km, harboured different sets of taxa. The same held true for the leaf-associated communities at different forest sites across Germany

and the zebra mussel-associated communities in different river systems (Extended Data Fig. 2), which have different biogeographic affinities[26].

**Natural DNA samplers reveal temporal biodiversity change**
We analysed temporal patterns of α-diversity (local OTU richness), temporal β-diversity (changes in community composition over time), spatial β-diversity (biotic homogenization or differentiation

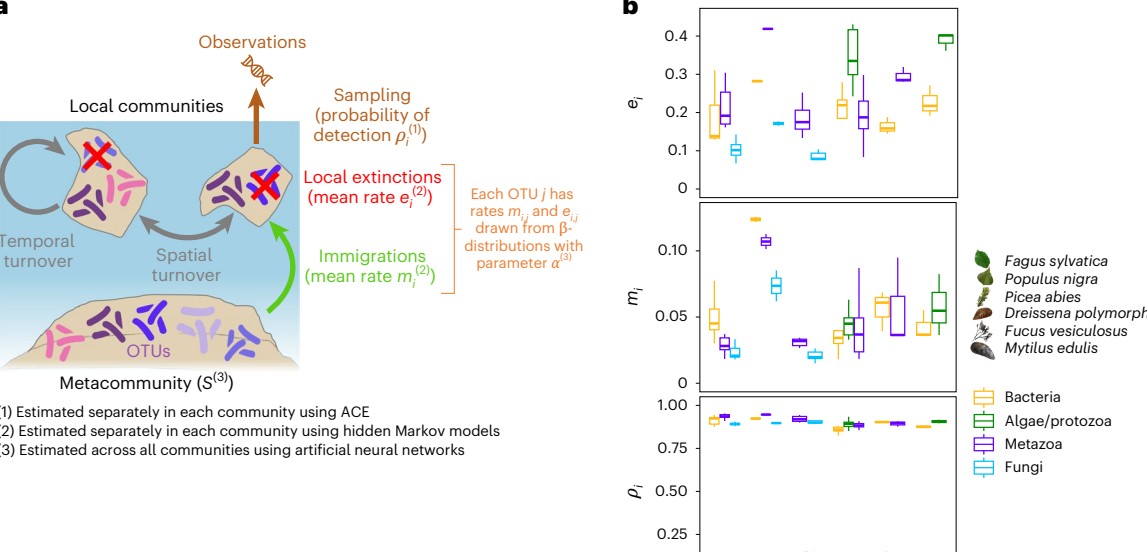

**Fig. 3 | A model of community ecology to assess temporal and spatial change in communities associated with natural sampler organisms from the ESB.** **a**, Schematic visualization of the non-neutral dynamic model for community assembly derived from the ETIB. **b**, Estimated parameters of extinction rate ($e_i$), immigration rate ($m_i$) and detection probability ($\rho_i$) in different taxonomic groups. Box plots show the median (centre line), the 25th and 75th percentiles (bounds of the box), and the minimum and maximum values (whiskers). No outliers were present in the data. For sample sizes underlying box plots see Data availability statement. Taxonomic groups are represented by different colours. Icons refer to sampled species.

of communities across sites) and γ-diversity (regional OTU richness). To evaluate the significance of the observed trends, we compared them with null expectations of community change in the absence of anthropogenic disturbances. These expectations were based on a dynamic model of community ecology that we developed, built upon the ETIB[17,24] (Fig. 3 and Methods). We validated this model using simulations (Extended Data Table 2 and Extended Data Fig. 3).

We identified no universal trend for α-diversity. Irrespective of ecosystems and taxonomic groups, richness remained stable, increased or declined (Fig. 4a,e, Extended Data Fig. 4a and Extended Data Table 3). A particularly pronounced drop in richness was found in marine prokaryotes. In contrast, richness strongly increased in limnic prokaryotes across the studied river systems. Interestingly, a reverse pattern was found for aquatic microeukaryotes, which showed α-diversity decreases in limnic habitats, but increases in marine sites (Fig. 4a,e). A more consistent pattern was found in terrestrial ecosystems, where most studied communities showed a slight increase of α-diversity across sites. This included a slight, albeit significant, increase in richness for some terrestrial arthropod communities (Fig. 4a,e and Extended Data Fig. 4a).

In contrast to α-diversity, a clear universal trend was found for changes in community composition over time (temporal β-diversity). This trend significantly deviated from the null expectations in all studied communities. We found considerable local extinctions of OTUs in all communities, which, however, were countered by the immigration of other taxa, leading to an out-of-equilibrium dynamic with a stronger-than-expected change in community composition (Fig. 4b,f, Extended Data Figs. 4b and 5a,c and Extended Data Table 3). Slightly distinct temporal dynamics of compositional change were observed across different taxonomic groups (Fig. 3b). In metazoans and microeukaryotes, very high rates of local extinction, combined with high rates of immigration, led to OTU-poor, rapidly changing communities. Bacterial communities were also characterized by high immigration rates, but had lower local extinction rates in comparison with metazoans and microeukaryotes. This resulted in more diverse, dynamic communities. In contrast, lower extinction and immigration rates

observed in fungi (Fig. 3b) generated slower compositional change compared with other communities, but still faster than expected. The observed changes were gradual in all communities: no abrupt breaks of community composition were detected (Fig. 4b,f and Extended Data Figs. 4b and 5a,c). Also, the compositional change did not affect higher taxonomic ranks: the temporal composition of phyla, classes and orders associated with the natural samplers remained remarkably stable (Fig. 2). The changes in community composition can be seen at the level of thousands of individual OTUs within our data. Each sampler type and data set showed replacements of various OTUs, with immigrations and local extinctions approximately balanced (Extended Data Fig. 6). We also detected various novel colonizers, among them typical invasive species like the Pacific oyster (*Crassostrea gigas*) or plant pathogens infesting trees across Germany (Supplementary Data 4). At the same time, declines in the occurrence of several taxa are evident, for example, the common periwinkle (*Littorina littorea*) in marine ecosystems and the green silver-lines (*Pseudoips prasinana*) in forest ecosystems.

We then assessed the evidence of biotic homogenization by exploring patterns of spatial β-diversity. Similar to α-diversity, no universal trend across ecosystems and taxonomic groups was evident (Fig. 4c,g, Extended Data Figs. 4c and 5b,d and Extended Data Table 3). Most aquatic communities did not show a clear trend of spatial homogenization over time. Some became even more differentiated across space, for example, marine metazoans. The increasing spatial heterogeneity was especially evident in prokaryotes and microeukaryotes across limnic sites (Fig. 4c,g and Extended Data Fig. 5b,d). In contrast, a general pattern of homogenization across space was evident in terrestrial canopy ecosystems. Nearly all the taxonomic groups, from prokaryotes to fungi and arthropods, showed a (significant) spatial homogenization of communities over time in the terrestrial samples (Fig. 4c,g and Extended Data Figs. 4c and 5b,d). As in the temporal β-diversity, the observed change was gradual rather than abrupt.

We found no universal pattern of γ-diversity across ecosystems and different taxonomic groups (Fig. 4d,h, Extended Data Fig. 4d and Extended Data Table 3). In general, regional richness (γ-diversity)

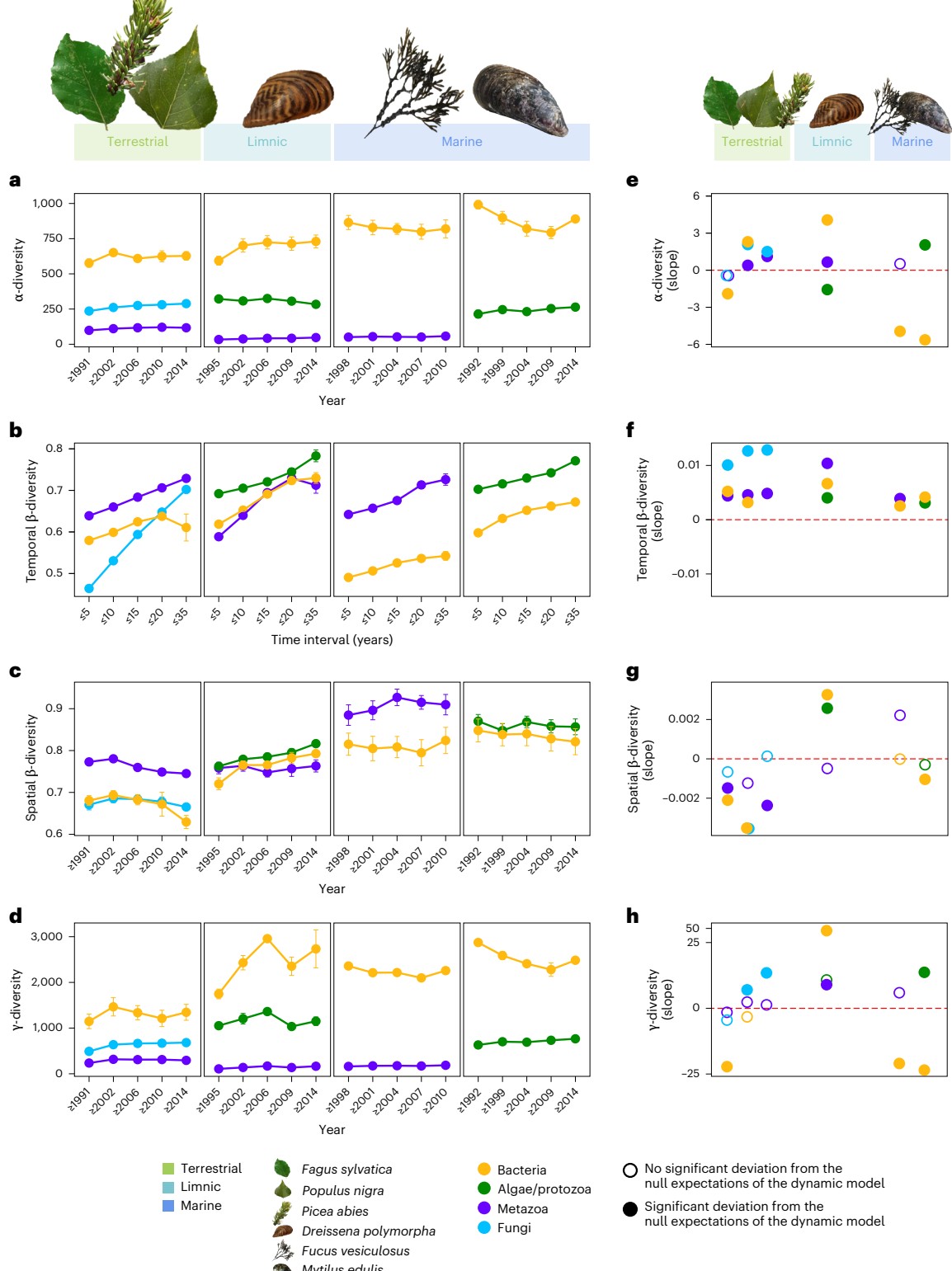

**Fig. 4 | Multidecadal trends of α-diversity, temporal and spatial β-diversity, and γ-diversity in communities across the tree of life associated with natural DNA sampler organisms from the ESB. a**, Trends of OTU richness (α-diversity) of the associated communities from 1991 to 2021. **b**, Temporal changes in community compositions (β-diversity measured using Jaccard distance) as a function of the time interval (in years) between samples from the same sampling site. **c**, Trends in spatial β-diversity (degree of dissimilarity in community composition between different sampling locations, measured using Jaccard distance) of the associated communities from 1991 to 2021. Results for temporal and spatial β-diversity are qualitatively similar when using only the turnover component of Jaccard distance for β-diversity

(Extended Data Fig. 5). **d**, Bootstrap estimates of regional diversity (γ-diversity) for the associated communities from 1991 to 2021. All diversity indices were summarized as mean with standard error bars across sampling locations and/or time windows (and for terrestrial samples across tree species). For a more detailed visualization see Extended Data Fig. 4; for sample sizes underlying means with error bars see Data availability statement. **e–h**, Diversity trends from **a** (**e**), **b** (**f**), **c** (**g**) and **d** (**h**) reduced to their respective slopes. Filled circles indicate significant departures from the null expectations generated through the dynamic model for community assembly, suggesting an out-of-equilibrium dynamic. Different ecosystems and taxonomic groups are represented by different colours. Icons refer to sampled species.

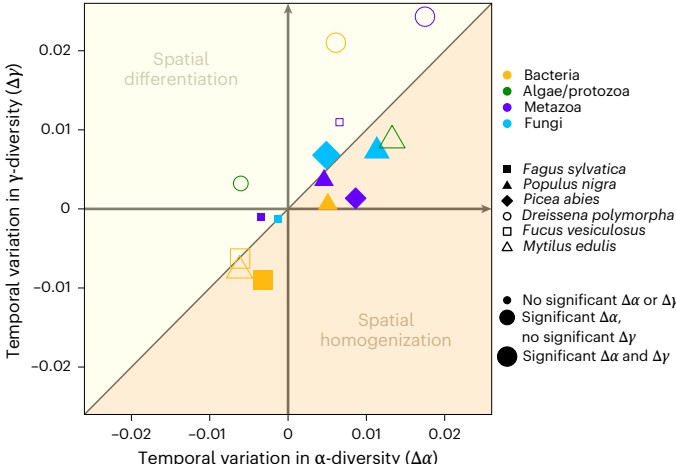

**Fig. 5 | Temporal variation of γ-diversity (regional richness) as a function of the temporal variation of α-diversity (local richness).** Following the conceptual framework of Blowes et al.[21], $\Delta\alpha < \Delta\gamma$ indicates spatial differentiation of community composition, whereas $\Delta\alpha > \Delta\gamma$ corresponds to spatial homogenization. The significance of the multidecadal variations of α- and γ-diversities of different communities associated with natural DNA sampler organisms from the ESB was assessed by measuring deviations from null expectations generated using the dynamic model for community assembly. Taxonomic groups are represented by different colours. Shapes refer to sampled species with their size indicating the significance level of $\Delta\alpha$ and $\Delta\gamma$.

trends roughly reflect local richness (α-diversity) trends, for example, with marine prokaryotes showing a strong decline, while a pronounced increase was found for limnic prokaryotes.

Finally, we explored the spatial variation of community composition within the conceptual framework of Blowes et al.[21]. This confirmed that most of the studied communities underwent a biotic homogenization (9/15 with $\Delta\alpha > \Delta\gamma$), while others (6/15 with $\Delta\alpha < \Delta\gamma$) experienced spatial differentiation (Fig. 5). Spatial differentiation was particularly significant in prokaryotes and metazoans in limnic sites and in fungi associated with Norway spruce, probably linked to the immigration of low-occupancy taxa across different sites ($0 < \Delta\alpha < \Delta\gamma$; Fig. 5). For instance, a bryozoan *Paludicella* OTU newly colonizing four out of nine sites in German river systems demonstrates this pattern of spatial differentiation. In contrast, spatial homogenization occurred in most terrestrial and marine communities associated with natural samplers. Yet, these communities exhibited variations in the type of homogenization. In most bacterial communities, the homogenization was owing to the local extinction of low-occupancy taxa at the different sites ($\Delta\gamma < \Delta\alpha < 0$; Fig. 5). This is exemplified by the disappearance of a *Wenyingzhuangia* (Bacteroidetes) OTU, which was unique to one of the North Sea bladderwrack sites, or the loss of an endophytic *Gluconacetobacter* (Pseudomonadota) OTU at the few beech sites previously inhabited by the genus. In other communities, homogenization can rather be linked to the widespread immigration of high-occupancy taxa ($0 < \Delta\gamma < \Delta\alpha$; Fig. 5), probably corresponding to invasive species or pathogens like *Taphrina*, a fungus newly infesting both our poplar sites (Supplementary Data 4).

## Discussion

Standardized time series data are critical to understand biodiversity change in the Anthropocene[5,6,17]. These data, however, are lacking for most taxa and ecosystems[5,7–9]. Here we show that archived natural samplers provide biodiversity time series data of unprecedented standardization across ecosystems and the tree of life[10,13].

Biodiversity decline is often assumed to be driven by local losses of α-diversity[8,16,20,23,27,28]. However, our data do not support universal declines of α-diversity, refuting our first hypothesis. Most ecosystems

did not lose diversity across individual sites; in fact, many communities even showed an increasing α-diversity. For instance, the analysis of archived leaf samples confirms the findings by Krehenwinkel et al.[13], suggesting that forest canopy arthropods are not affected by pronounced insect decline at local scales.

Instead of universal declines of α-diversity, we observe a widespread taxonomic replacement across all studied taxonomic groups. Our data indicate that thousands of species have disappeared from Germany's aquatic and terrestrial ecosystems in the past decades. The local extinctions of species, however, are countered by the immigration of novel, possibly better adapted taxa, supporting our second hypothesis[14,17]. This corresponds to an out-of-equilibrium dynamic in both aquatic and terrestrial ecosystems in Germany, potentially driven by anthropogenic disturbances[29]. The compositional change appears gradual across all taxonomic groups and ecosystems, following our hypothesis (2a). Also, taxonomic replacement mostly affects rather closely related taxa, with no distinct changes in higher-level taxonomic groups. This suggests that the studied communities have probably retained their ecological make-up and taxonomic replacements are functionally redundant[30–32]. Considering this background, we reject the hypothesis of rapid state shifts in communities that have reached tipping points (2b)[19,22]. The observation of a gradual compositional turnover as a predominant pattern of biodiversity change in the Anthropocene is well supported by other recent work[5,17,33,34]. The relatively similar rates of compositional change across all sampler organisms suggest a common driver, for example, climate change[2].

Our results highlight another facet of environmental change: gradual biotic homogenization across space[21,28,35], especially in terrestrial and marine ecosystems. In bacterial communities associated with beech leaves and marine samples, the observed homogenization is probably due to the local extinction of rare taxa[21]. In contrast, other communities showing biotic homogenization may have acquired novel widespread taxa. Such widespread generalist species frequently benefit from environmental change and replace more locally adapted, specialized species[21,36–38]. Consequently, the results support our third hypothesis for marine and especially terrestrial ecosystems. Interestingly, limnic ecosystems show contrasting patterns. The increasing geographic differentiation of aquatic communities in Germany over time might be caused by the ongoing spreading of invasive taxa and/or the immigration of novel taxa. The latter may be facilitated by artificial links between river systems, for example, the Rhine–Main–Danube Canal or shipping[39,40]. Results involving γ-diversity should, however, be interpreted with caution due to few sampling sites and their heterogeneous availability at different times (Supplementary Data 1) in some datasets. Also, some communities shifted from an upward to a downward diversity trend (or vice versa) over time. These limitations may have led to the slight inconsistencies in spatial β-diversity trends (Fig. 4g) and the framework of Blowes et al.[21] (Fig. 5) in limnic and beech-associated metazoans. Yet, the overall picture of either spatio-temporal homogenization or differentiation across different taxonomic groups and ecosystems is consistent across both approaches and supported by recent work on time series data from all across the globe[21,41].

The majority of the increasing or declining taxa that we observe are highly cryptic and rarely detected by monitoring programmes, which focus on prominent taxa like plants[42] or vertebrates[43]. Yet many of the taxa we observe represent critical elements in food webs, for example, phytoplankton in aquatic ecosystems or arthropods in tree canopies[44–46]. Future work should further explore the ecological role of the recovered taxa in their respective ecosystems and food webs. ESB samples can serve as an important early warning system for both the decline of local species and the emergence of pathogens or problematic invaders[13,14].

We here focused on ecosystems across Germany, but future work on natural sampler organisms from other ESBs may also enable the study of biodiversity change at broader geographical scales. ESBs

have been established in various countries[47], holding the potential to explore global patterns of biodiversity change with natural samplers. Moreover, recent work has shown that DNA in natural samplers can show a remarkable temporal stability, even without long-term cold storage[48]. For example, community-level DNA has been successfully isolated from plant herbarium specimens[49], which opens up natural history collections as a promising source for future studies on biodiversity change. Natural sampler organisms thus hold great promise to provide the long-term global data that are so direly needed to understand and mitigate biodiversity decline.

## Methods
### Specimen bank data
This research complies with all relevant ethical regulations. All study protocols are approved by the German Environment Agency.

The German Environmental Specimen Bank (ESB) has been in operation since the early 1980s. The ESB collects samples of indicator species from various terrestrial and aquatic ecosystems. These species serve as accumulators of environmental chemicals and provide a detailed image of pollution and ecosystem health. To make pollution analysis comparable between years, ESB samples are collected according to highly standardized protocols. Samples are taken at the same time of the year, at identical sites and using identical protocols. Collection is done using sterile equipment to avoid carryover of even trace amounts of pollutants between samples. To ensure preservation of unstable chemical compounds, the samples are stored over liquid nitrogen after collection and for the long term, halting all chemical and biological degradation. To acquire an integrative view of pollution in an ecosystem, ESB samples are large, each one including hundreds to thousands of specimens or tissue compartments (leaves in case of trees)[10,50–55]. Each sample is cryomilled to a fine powder of a grain size of 200 μm, thoroughly homogenizing all traces of chemicals[56].

Recent work shows that ESBs are ideal for studies on biodiversity change. ESB indicator species can be considered natural community DNA samplers, which preserve an imprint of the surrounding biological community at the time of sampling. The highly standardized and contamination-free sampling and sample processing conditions, coupled with storage at ultra-low temperatures, make ESB samples perfectly suited for metabarcoding. The cryomilling of large ESB samples also guarantees an even distribution of community DNA traces among the sample and breaks open cell walls of various microorganisms, whose DNA is then uniformly released. Previous studies have already extensively tested and highlighted the suitability of different ESB samples for retrospective biodiversity monitoring[12–14]. Here, we use four different types of ESB samples from terrestrial, limnic and marine habitats as natural community DNA samplers to measure biodiversity change across four decades.

**Tree leaves.** The ESB collects leaves from three common tree species in Germany, namely European beech (*Fagus sylvatica*), Norway spruce (*Picea abies*) and Lombardy poplar (*Populus nigra*). The leaves are collected once annually or biannually and serve as samplers for aerial pollutants deposited on the leaves[51,52,54]. ESB leaf samples are collected from different forest ecosystem types, spanning a land use gradient from core zones of national parks to timber forests, forests neighbouring agricultural sites, and urban parks. Sample series from nine sites were included in this study, starting from 1985. Each sample contains hundreds of leaves from at least 15 individual trees, milled to a fine powder[51,52,54,56]. These samples contain DNA traces of all organisms that interacted with the tree canopy at the time of collection[13]. Here, we characterize communities of canopy-associated arthropods, fungi and bacteria. The results for arthropod diversity shown in our study represent a novel dataset compared with Krehenwinkel et al.[13], including additional and longer time series and an improved DNA extraction protocol to deal with polymerase chain reaction (PCR) inhibitors. Only

the poplar datasets as well as the 16S ribosomal DNA (rDNA) datasets were taken from the original dataset by Krehenwinkel et al.[13].

**Bladderwrack (*Fucus vesiculosus*).** This macroalgae is widespread along the European coastline, where it makes up a substantial part of the biomass. Marine pollutants are enriched in the tissue of the algae, making it an ideal sentinel species for pollution monitoring[53]. Three sites have been sampled annually or biannually for bladderwrack thalli beginning in 1985. ESB samples from two North Sea sites are collected at intervals of two months, six times a year, and then merged into a pooled annual sample. The third site at the Baltic Sea is sampled twice a year. Bladderwrack is a critical species in coastal ecosystems, providing a habitat for countless taxa. All these taxa leave detectable DNA traces in the ESB sample[12]. Here we characterize communities of animals and bacteria that interacted with the bladderwrack.

**Blue mussels (*Mytilus edulis*).** This is the most common mussel in coastal ecosystems of northern and central Europe. Blue mussels constantly filter the water column for planktonic organisms and organic particles. In doing so, they enrich pollutants in their tissue, making them an excellent sentinel species for pollution monitoring[55]. The ESB has collected blue mussels at three coastal sites in Germany since 1985. The mussel's entire soft tissue including respiratory water is used for the ESB sample. Annual or biannual samples of hundreds of mussels are compiled from six sampling events at the North Sea and two at the Baltic Sea. With each mussel filtering roughly 1 litre of water per hour, these samples contain a comprehensive imprint of the annual planktonic biodiversity at the sampling site[12]. Here we characterize communities of eukaryotic plankton and mussel-associated bacteria.

**Zebra mussels (*Dreissena polymorpha*).** The limnic zebra mussel is an invasive species from the Black Sea region, which has colonized nearly all major rivers of Germany since the 1960s[57]. Like the blue mussel, zebra mussels are highly efficient filter feeders. Since the 1990s, zebra mussels are reared by the ESB on special plate stacks, which are then placed in four major German rivers for about one year, allowing the mussels to accumulate pollutants in their tissue. The mussels are then collected from the plate stacks, immediately deep-frozen and a sample of soft tissue including respiratory water is compiled from thousands of mussels[50]. The samples from nine sites used here provide an overview of limnic biodiversity from major rivers of Germany[12]. Here we characterize communities of animals, eukaryotic plankton and mussel-associated bacteria.

### Laboratory workflow and sequence processing
Samples were processed as described in refs. 13,14. Work steps were performed on clean benches to avoid carryover and cross-contamination. We isolated DNA from 200 mg of homogenate from each sample. This amount was shown to be sufficient to recover the sample-associated diversity in ESB leaf samples[13] and is assumed to apply to all ESB sample types due to the identical grinding process[56]. DNA was extracted in one or two replicates depending on the sample type (Supplementary Data 5) using a CTAB protocol (OPS Diagnostics), which proved best suited to extract high-purity DNA from these sample types. The DNA extracts were then amplified for different DNA metabarcode markers to enrich various taxonomic groups from the samples (for a list of metabarcode markers and PCR conditions see Supplementary Data 5). PCR was performed using the Qiagen Multiplex PCR Kit in 10-μl volumes according to the manufacturer's protocols. Primers were chosen to amplify the associated community, but not the ESB indicator species itself, whose DNA dominates the extract. To characterize bacterial communities (all sample types), we amplified the V1 or V5–7 region of 16S rDNA[58–60]. For the bladderwrack and tree samples, mitochondrial and chloroplast DNA will be greatly overrepresented over bacterial DNA in the samples. We thus used primers that exclude chloroplast and mitochondrial

amplification for these samples. This may also result in slightly lower species numbers of bacteria recovered, but should not affect the community composition within sample types. For terrestrial arthropods (tree leaf samples), we used a mitochondrial cytochrome oxidase I (COI) marker[48]. For fungi (tree leaf samples) we used the ITS1 region of the nuclear ribosomal cluster[61,62]. The variable V9 region of nuclear 18S rDNA was targeted to characterize communities of aquatic animals and eukaryotic plankton (bladderwrack and mussel samples[12]; for primer details, see Supplementary Data 5). PCR success was checked on 1.5% agarose gels, and the PCR products were then amplified in another round of PCR to add Illumina TruSeq adaptors and unique combinations of dual indexes[63] (Supplementary Data 5). All final libraries were pooled in approximately equimolar amounts, cleaned of leftover primers using 1X AMPure beads XP (Beckman Coulter) and then sequenced on an Illumina MiSeq using paired-end sequencing with 500-cycle V2 and 600-cycle V3 kits. To ensure reproducibility of our data and to recover rare species, we ran several PCR replicates for every sample, which were indexed and sequenced separately. The number of PCR replicates was adapted based on sample type and marker, and varied between three and six (Supplementary Data 5). Blank DNA extractions were included in every batch of extractions, and non-template control PCRs were run alongside all PCR reactions. All controls were sequenced along with the samples to provide a baseline for carryover or cross-contamination during processing.

Forward and reverse reads were merged using PEAR[64] with a minimum quality of 20 and a minimum overlap of 50 base pairs (bp). The merged reads were then quality-filtered by limiting the number of expected errors in a sequence to 1 (ref. 65) and transformed to FASTA format using USEARCH[66]. Primer sequences were trimmed off using Unix scripts. Long 18S rDNA amplicons (~350 bp) of limnic metazoan and microeukaryotic plankton generated from zebra mussels were trimmed to match the corresponding short amplicon of ~150 bp. As both metazoan and phytoplankton amplicons span exactly the same nuclear 18S rDNA region, all sequences were combined into one file. Likewise, reads generated from the three tree species were saved to one file for each marker. After trimming, the resulting file for each marker and sample type was dereplicated and clustered into zero-radius OTUs using the USEARCH pipeline. OTU tables were built for each sample type and marker, also using USEARCH. Taxonomy was annotated using blast2taxonomy script v1.4.2.[67] after BLAST searching[68] against the entire National Center for Biotechnology Information GenBank database for 18S rDNA and COI with a maximum number of ten target sequences. The SILVA database[69] was used for annotating 16S rDNA sequences, and the UNITE database[70] for fungal ITS1. The FungalTraits database[71] was used for the functional annotation of fungi. Taxonomic assignments based on BLAST hits with a base pair length of less than 80% of the amplicon length and/or less than 85% sequence identity were removed. We excluded all taxa except bacteria, algae/protozoa, metazoa or fungi from the respective datasets. Some fungal and bacterial DNA in the samples may also stem from small metazoans associated with the samples. These may bias the recovered diversity patterns from these microorganisms. To ensure independence between recovered biodiversity trends of different taxonomic groups, we functionally annotated fungi and bacteria in our terrestrial dataset and excluded those associated with arthropods. Results for the calculated diversity trends did not differ from those presented in Fig. 4. Furthermore, we used the FungalTraits[71] and PLaBAse[72,73] databases to check for the proportion of plant-associated fungi and bacteria in our terrestrial dataset. Of all OTUs with a genus-level annotation and a match in the respective reference database, we verified 34% (fungi) and 76% (bacteria) as plant-associated taxa.

Per OTU and sample replicate, we removed all reads below 3, as this is the read carryover between samples commonly observed in our workflows. The OTU tables were checked for contamination using negative controls by excluding OTUs present in the negative controls from the dataset (typical laboratory contaminants). We only detected negligible contamination in the samples. PCR replicates were merged and all datasets were checked for sufficient sequencing depth and sampling size (Extended Data Fig. 1; for resulting sampling and OTU count as well as number of phyla and orders see Supplementary Data 2 'Cleaned dataset').

## Statistical model and analyses of community diversity

For each sample type, we (1) only selected sites that were sampled for at least 5 years; (2) only kept sampling years represented by at least 50% of the sites; and (3) removed years that were isolated from the others (>2 years). We also removed samples with low read coverage (less than 50% of the median number of reads). Finally, because OTU read abundances from metabarcoding datasets are subject to many biases[74], we converted OTU abundances into binary presence/absence data and only analysed trends in terms of OTU occurrence. To limit cross-contamination, we considered an OTU as present in a sample if it represented at least 0.01% of the total reads. A total of 537 samples were included in the analysis (for resulting sampling and OTU count as well as number of phyla and orders per dataset; see Supplementary Data 2 'Filtered dataset (model)' and Extended Data Table 1).

We measured community diversity trends in four different ways. First, in each community, in each year, we computed the α-diversity using the OTU richness as a measure of local diversity at a given time. Second, we computed β-diversities between pairs of communities (1) sampled at the same site in different years; or (2) sampled at different sites in the same year. Option (1) gives an idea of changes in community composition over time (temporal β-diversity), whereas (2) indicates changes in community composition across space (spatial β-diversity). We measured β-diversities using the full Jaccard distances (turnover and nestedness) and the turnover component of the Jaccard distances alone (R package betapart[75]). Last, for each dataset, each year, γ-diversity (regional diversity) was computed using bootstrapping with the specpool function (R package vegan[76]).

To identify temporal trends in these diversity indices, temporal models were fitted using mixed linear models accounting for the temporal autocorrelation between sampling years and the effect of the different sampling sites. We used the lme function (R package nlme[77]) with the corAR1 temporal correlation and the different sites as random effects. We fitted these temporal models with either the α-diversities, the temporal or spatial β-diversities, or the γ-diversities as response variables.

The significance of the observed trends was evaluated by comparing them with null expectations of community changes. Changes in community composition occur as a result of immigration and local extinction events, which are influenced by neutral and/or niche factors. This generates a dynamic equilibrium with ever-changing communities, even in the absence of any kind of disturbance. Following ref. 17, we built upon the ETIB[24] to set up a dynamic model for community assembly that generates null expectations of diversity trends in the absence of disturbance (Fig. 3a). The ETIB is a lineage-based model of species colonization of a local community (the island) from a metacommunity (the continent). In its simplest form[24], at each time step, it assumes that each OTU has a probability $m_i$ to migrate from the metacommunity to the local community $i$, and once settled in the community, each OTU has a probability $e_i$ to go extinct. The number of new immigration events (that is, of OTUs not already in the community $i$) per time step is given by $m_i(S - s_i)$, where $S$ is the total number of OTUs in the metacommunity and $s_i$ is the number of OTUs already present in community $i$; it declines as the number of OTUs in the community increases. The number of local extinction events per time step, given by $e_i s_i$, increases with the number of OTUs in the community. An equilibrium is reached when the number of immigration events per time step equals the number of extinction events, that is, $m_i(S - s_i) = e_i s_i$. The equilibrium number of OTUs in the community is given by $s_i = m_i S / (m_i + e_i)$. This simple form of

ETIB implies a linear decrease (respectively, increase) of the number of new immigration events (respectively, extinction events) per unit of time with the number of settled OTUs in the local community. It assumes that all OTUs have the same probabilities to migrate or go extinct (neutrality), and that these probabilities do not depend on the number of OTUs in the community, implying that there is a negligible influence of interspecific competition on immigration and extinction. It thus applies best to communities that are far from carrying capacity. This model has the advantage of being very straightforward to simulate using a simple discrete-time Markov chain[78].

Given the incomplete sampling of communities typically achieved with metabarcoding techniques, we assumed that each OTU present in community $i$ is observed at each time step with a fixed probability $\rho_i$. This extra parameter can be handled using hidden Markov models. We assumed that the rates $m_i$, $s_i$ and $\rho_i$ vary from one community to the other due to various extrinsic and intrinsic factors (for example, distance to the metacommunity, community size and environmental factors).

Assuming neutrality, that is, that all OTUs have the same immigration and extinction rates, is a strong assumption often proved to be unrealistic[79]. We therefore relaxed the assumption of neutrality. In our non-neutral model, we assumed that immigration (respectively, extinction) rates for each OTU are sampled from a beta distribution with parameters $\alpha$ and $\alpha(1 - m_i)/m_i$ (respectively, $\alpha(1 - e_i)/e_i$). The rates are therefore sampled around $m_i$ (respectively, $e_i$) with a variance that is inversely proportional to $\alpha$: a large $\alpha$ corresponds to scenarios of neutrality whereas $\alpha$ closer to 0 indicates that immigration and extinction rates are very different across OTUs. While each OTU has specific immigration and extinction rates in each community, we assumed that the ranks of the OTUs in terms of immigration and extinction rates are conserved across communities (that is, an OTU with a low extinction rate in community $i$ compared with other OTUs also has a low extinction rate in community $j$). We thus obtained a non-neutral model derived from the ETIB, which assumes that the presence of an OTU in a community results from the balance between immigration and local extinction, and that each OTU is characterized by specific rates of immigration and local extinction centred around the average rates. At equilibrium, some OTUs are more likely to be present in the community (for example, OTUs with higher immigration and lower extinction rates).

Instead of testing different parameter values chosen a priori[17], we implemented an inference strategy to adjust the model parameters to the empirical data using a sequential technique (Fig. 3a). First, in each community, the sampling fraction $\rho_i$ was inferred using ACE (R package vegan[80]). Second, we estimated the average rates $m_i$ and $e_i$ by fitting the neutral model to each community using a hidden Markov model (R package seqHMM[81]). We used these estimates as community-specific estimates of the average rates of immigration and extinction of the non-neutral model. Third, given $\rho_i$, $m_i$ and $e_i$, we used simulation-based inferences using artificial neural networks to estimate the parameters $S$ and $\alpha$. We generated a large number of simulated datasets by sampling $S$ and $\log \alpha$ from uniform prior distributions and simulating the corresponding non-neutral model of community assembly, and for each of these simulations, we recorded α-, spatial and temporal β-, and γ-diversities through time across all the sampled sites. We specifically incorporated subsampling into the simulations (with probability $\rho_i$), such that not all OTUs present in the local communities are observed, mimicking the detection bias of metabarcoding: simulated α-, spatial and temporal β-, and γ-diversities are therefore directly comparable to empirical diversites. For $S$, we used a uniform prior distribution between the number of observed OTUs and three times the estimated γ-diversity; for $\log \alpha$, we used a uniform prior between 1 and 5. We started the simulations at year 1500 (providing ample time to ensure they reach equilibrium) with a random community composition at each site (each OTU has a probability $s_i/S$ to be initially present in the community, where $s_i$ is the theoretical number of species at equilibrium in the

neutral model, given $S$, $m_i$ and $e_i$: $s_i = m_i S/(m_i + e_i)$). Next, we simulated community composition over time in each site until 2023, sampled the communities according to $\rho_i$ and recorded community composition through time for the years of sampling. We then computed for each simulation the α-, β- and γ-diversities: we used the same methods and sampling scheme as for empirical data (the number of sampling sites varied through time) to obtain comparable measures of α-, spatial and temporal β- as well as γ-diversities. We trained an artificial neural network to estimate $S$ and $\log \alpha$ from time series of α-, β- and γ-diversities using the Python library Keras[82], with 100,000 simulations per dataset until reaching a sufficient predictive power. Once trained, the artificial neural network takes as input α-, β- and γ-diversities through time and outputs estimates of $S$ and $\log \alpha$. We used a neural network with three intermediate layers, containing 132, 64 and 32 neurons respectively, and with exponential linear unit (ELU) activation functions. We prevented overfitting by using a dropout of 0.5 at each intermediate layer. Input and output data were scaled between 0 and 1 before fitting, and the simulations were split between the training set (90%) and the test set (10%). Once validated (Extended Data Fig. 3), we finally applied the trained neural network to our empirical data, and obtained corresponding $S$ and $\log \alpha$ values.

Given the estimated $S$ and $\log \alpha$ values, we simulated our non-neutral model 1,000 times with these parameters to generate time series of α-, β- and γ-diversities under an equilibrium model of community assembly. We then compared empirical and simulated temporal trends and considered an empirical trend to be significant if it fell outside the central 95% of the distribution of the simulated trends. $P$ values were computed as the proportions of simulated trends that were higher or lower than the empirical trend. We interpreted significant deviations from the simulated equilibrium model of community assembly as indicative of out-of-equilibrium dynamics in the empirical data, potentially driven by anthropogenic disturbances.

We used simulations to test the validity of our approach that assesses the significance of the observed trends by comparing them to null expectations of community changes under the non-neutral dynamic model for community assembly. We simulated two scenarios: (1) a scenario without any disturbance; and (2) a scenario of biotic homogenization and regional diversity loss. To generate realistic simulations, we designed them using the number of sites, the total number of OTUs, the rates of immigrations and local extinctions, and the sampling probabilities estimated from the bacterial communities of beech (one of the largest datasets). In the first simulated scenario, we simply simulated community changes under the assumptions of our non-neutral dynamic model for community assembly—we therefore did not expect any deviations from the null expectations when applying our approach. Conversely, in the second simulated scenario, we simulated the regional extinctions of 10% of the OTUs and their replacement by widespread invasive OTUs across all communities—we therefore expected no variation of α-diversities, a decrease of spatial β-diversities (spatial homogenization) as well as a decrease of γ-diversities (regional diversity loss). We performed ten simulations for each scenario. When applying our approach, we correctly recovered the simulated scenario in almost all cases (Extended Data Table 2), confirming the validity of our approach.

To interpret the observed variations of community compositions, we used the conceptual framework of Blowes et al.[21]. For each sample type and taxonomic group, we plotted the temporal variation log γ-diversities as a function of the temporal variation of log α-diversities. By construction, $\Delta \alpha < \Delta \gamma$ indicates spatial differentiation of the community composition (increase of spatial β-diversities), whereas $\Delta \alpha > \Delta \gamma$ corresponds to spatial homogenization (decrease of spatial β-diversities).

Finally, we investigated the temporal trends of OTU occurrences, as some OTUs may be recent invaders and others may have gone extinct at the regional scale. For each sample type and taxonomic group, we

only looked at OTUs present in at least 10% of the samples and across at least two sites. For each OTU, we first tested whether its occurrence in the communities tended to vary through time. To do so, we fitted a generalized linear mixed model with a binomial response (presence/absence of the OTU in a community) and considered the sampling site as random effects.

## Reporting summary

Further information on research design is available in the Nature Portfolio Reporting Summary linked to this article.

## Data availability

The data that support the findings of this study are available via the Science Data Bank repository at https://doi.org/10.57760/sciencedb.13553 (ref. 83). Raw Illumina sequencing data are available in the European Nucleotide Archive repository under accession number PRJEB88877. Source data are provided with this paper.

## Code availability

R code for data analysis and the dynamic model is available via the Science Data Bank at https://doi.org/10.57760/sciencedb.13553 (ref. 83).

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

## Acknowledgements

We thank K. Fischer for assistance with laboratory work. We want to thank A. Frohböse-Körner from the German Environment Agency (Umweltbundesamt) for her assistance in getting access to metadata for the samples. This work was funded by departmental funds of the department of Biogeography at Trier University and by the TrendDNA project of the German Environment Agency. We want to especially thank the ESB project group at Trier University and the German Environment Agency for providing the time series samples.

## Author contributions

I.J., J.H., A.M., S.W., M.S., E.G., C.S., A.S., J.K. and N.S. performed the laboratory work and analysed the data. H.K., I.J., J.H., B.P.-L. and H.M. conceptualized the study, analysed the data and wrote the manuscript. B.P.-L. and H.M. developed the dynamic model for community ecology. S.K. performed the sequencing. M.P., R.K. and D.T. led the sampling. J.K. enabled access to the samples. All authors contributed to revising the final version of the manuscript.

## Funding

## Competing interests

The authors declare no competing interests.

## Additional information

**Extended data** is available for this paper at https://doi.org/10.1038/s41559-025-02812-6.

**Correspondence and requests for materials** should be addressed to Henrik Krehenwinkel.

¹Trier University, Trier, Germany. ²Institut de Biologie de l'ENS (IBENS), Ecole Normale Supérieure, CNRS, INSERM, Université PSL, Paris, France. ³Université de Toulouse, Toulouse INP, CNRS, IRD, CRBE, Toulouse, France. ⁴University of California, Berkeley, CA, USA. ⁵University of Cologne, Cologne, Germany. ⁶Umweltbundesamt, Berlin, Germany. ⁷These authors contributed equally: Isabelle Junk, Julian Hans, Benoît Perez-Lamarque. ✉e-mail: krehenwinkel@uni-trier.de

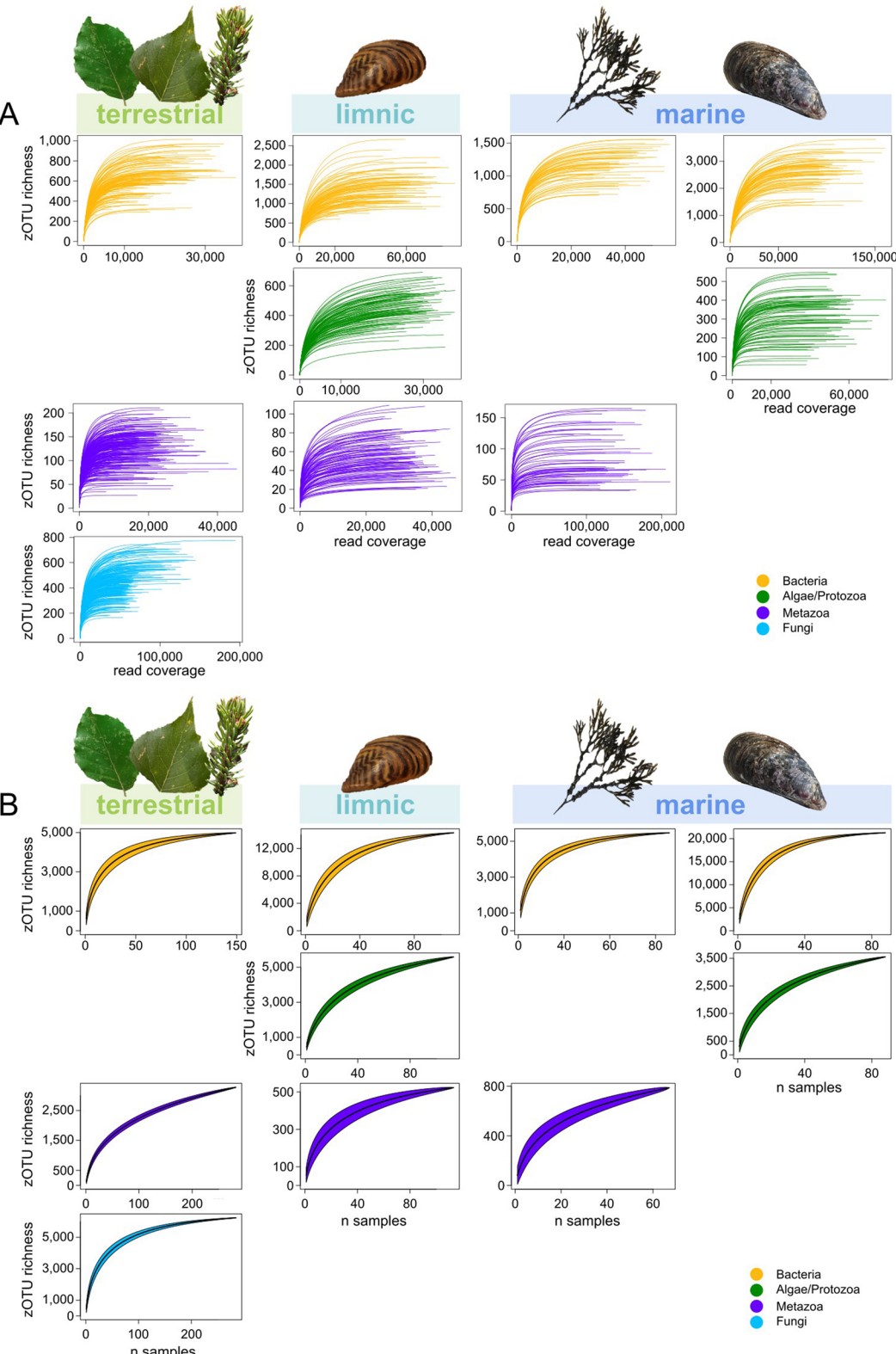

**Extended Data Fig. 1 | Rarefaction and accumulation curves depict the saturation of OTU richness for each sampling type. a)** Rarefaction curves showing OTU richness as a function of read coverage, with each curve representing an individual sample. **b)** Mean accumulation curves (central black line) and their standard deviation (colored area) from random permutations of the data illustrate the cumulative increase in OTU richness with each additional sample. Taxonomic groups are represented by different colors. Icons refer to sampled species.

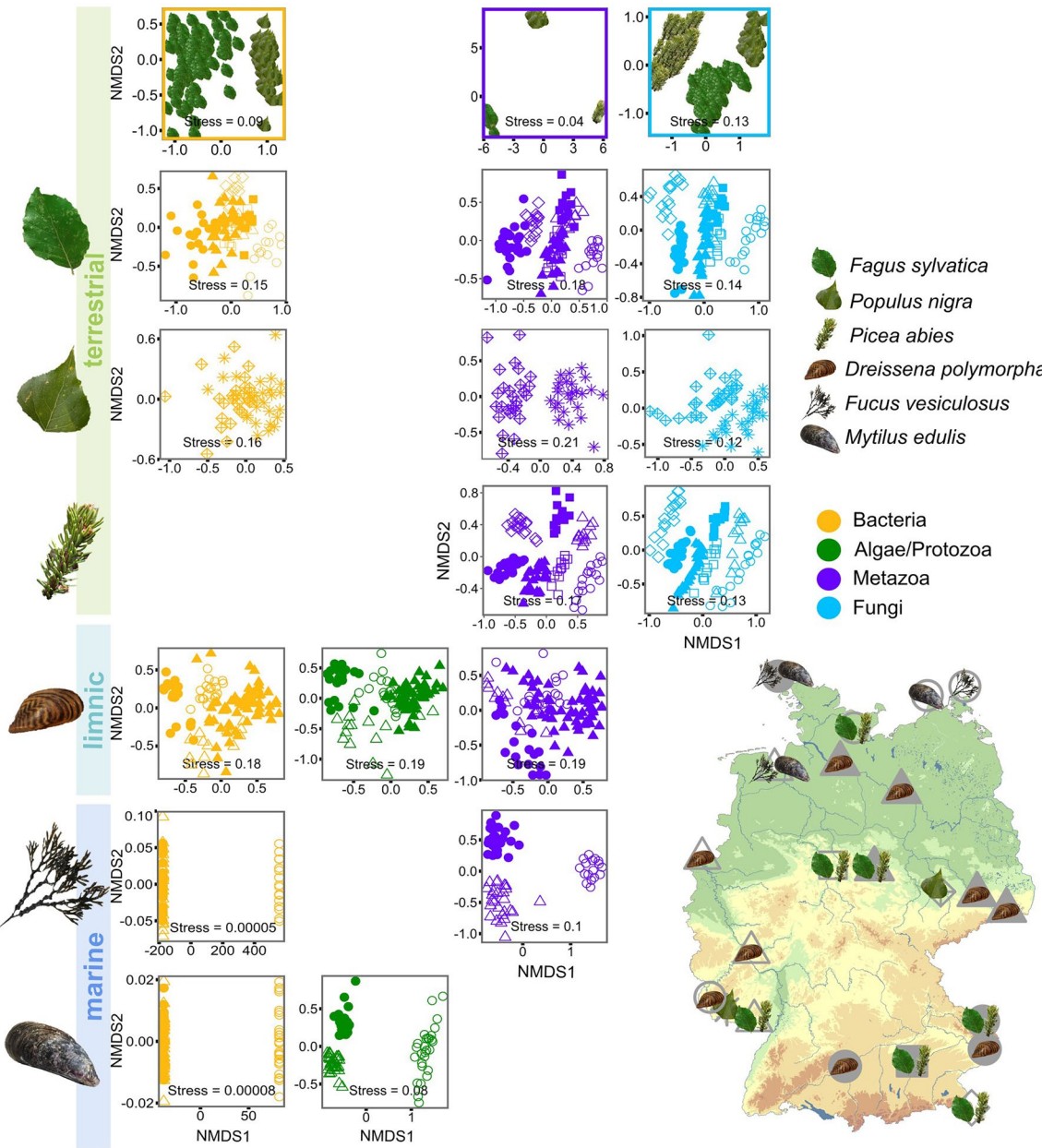

**Extended Data Fig. 2 | Map of sampling sites and non-metric multidimensional scaling (NMDS) plots illustrating distinct bacterial, microeukaryotic plankton, metazoan and fungal communities at different sampling sites within terrestrial, limnic and marine ecosystems.** The first row shows NMDS plots of Bacteria, Metazoa and Fungi associated with the leaves of three tree species - European beech, Norway spruce, and Lombardy poplar - where each tree species is represented by a distinct icon. Rows two through four illustrate the studied leaf-associated communities for each tree species separately. Rows five to seven show NMDS plots of Bacteria, Algae/Protozoa and Metazoa associated with limnic zebra mussels, as well as marine bladderwrack and blue mussel. Taxonomic groups are represented by colors. Icons refer to sampled species. Shapes denote the sampling location.

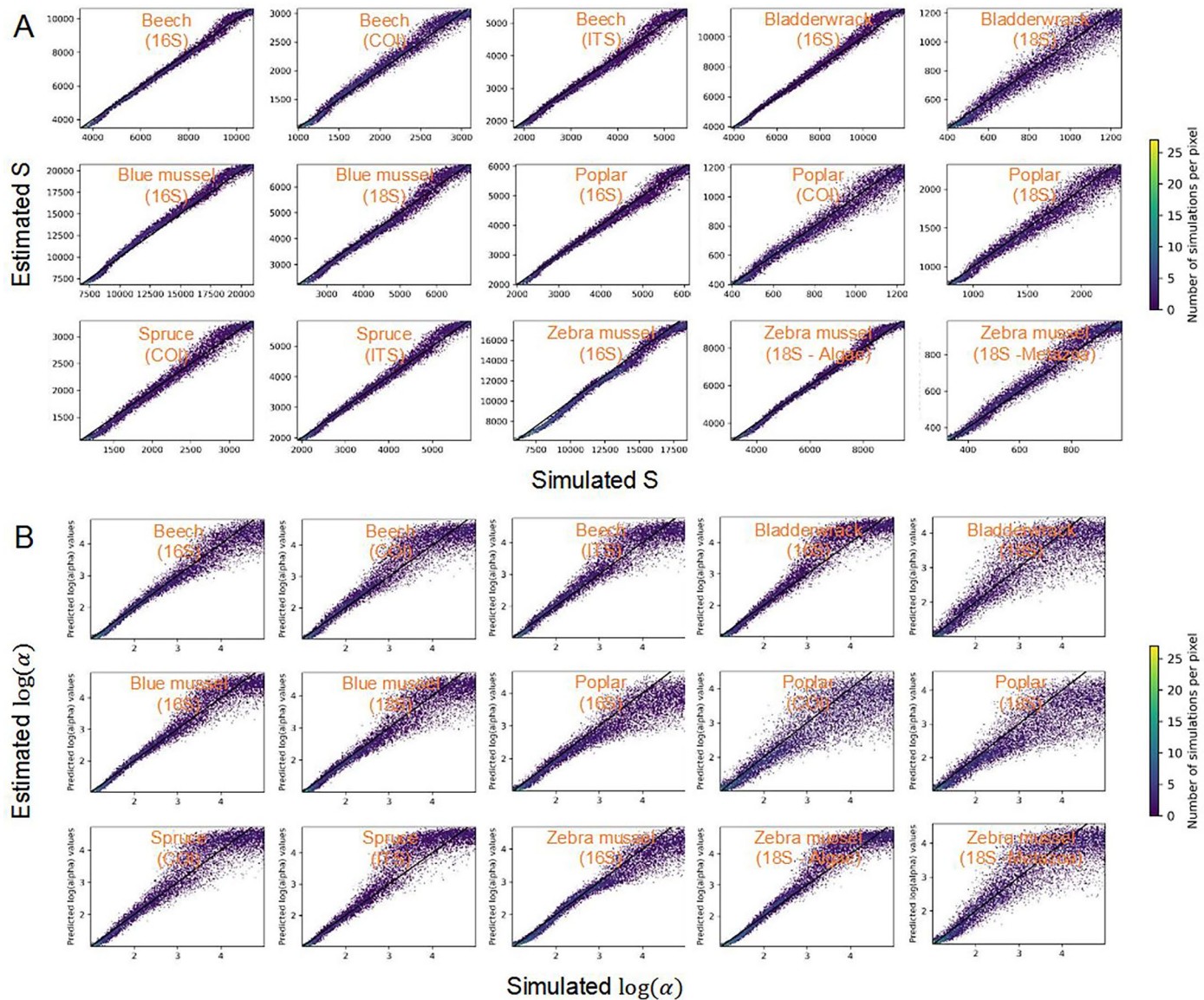

**Extended Data Fig. 3 | Validation of parameter estimations using the artificial neural networks for the dynamic model of community assembly. a)** Estimated values of the total number of OTUs in the metacommunity (S) as a function of the simulated one. **b)** Estimated values of parameters of the beta distribution (log(α)) as a function of the simulated ones. The black lines y = x represent perfect estimations.

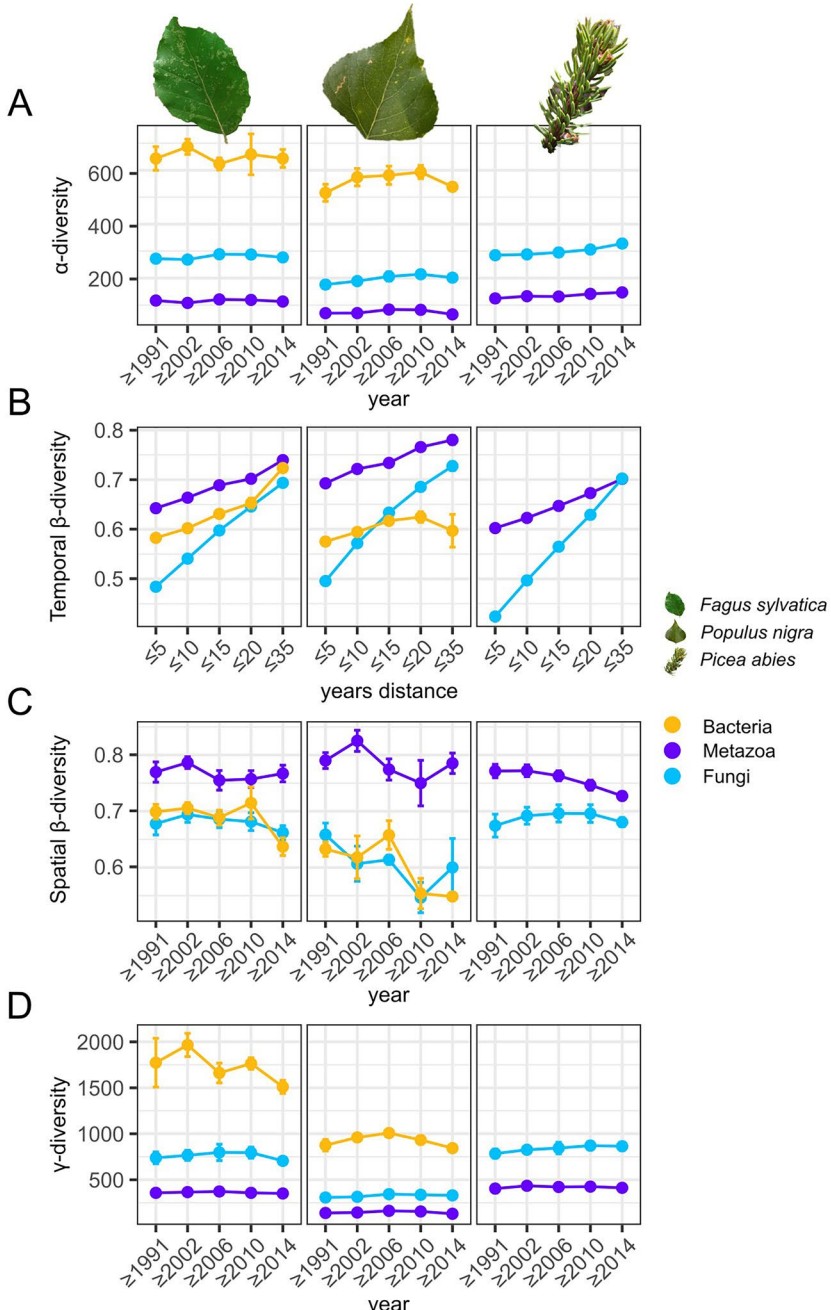

**Extended Data Fig. 4 | Multidecadal trends of α-diversity, temporal and spatial β-diversity and γ-diversity in communities across the tree of life associated with tree leaves, with each tree species presented separately.**
**a**) Trends of OTU richness (α-diversity) of the associated communities from 1991 to 2021. **b**) Temporal changes in community compositions (β-diversity measured using Jaccard distance) as a function of the time interval (in years) between samples from the same sampling site. **c**) Trends in spatial β-diversity (degree of dissimilarity in community composition between different sampling locations, measured using Jaccard distance) of the associated communities from 1991 to 2021. **d**) Bootstrap estimates of regional diversity (γ-diversity) for the associated communities from 1991 to 2021. All diversity indices are summarized as mean with standard error bars across sampling locations and/or time windows. For underlying sample sizes see data availability statement. Different taxonomic groups are represented by different colors. Icons refer to sampled species.

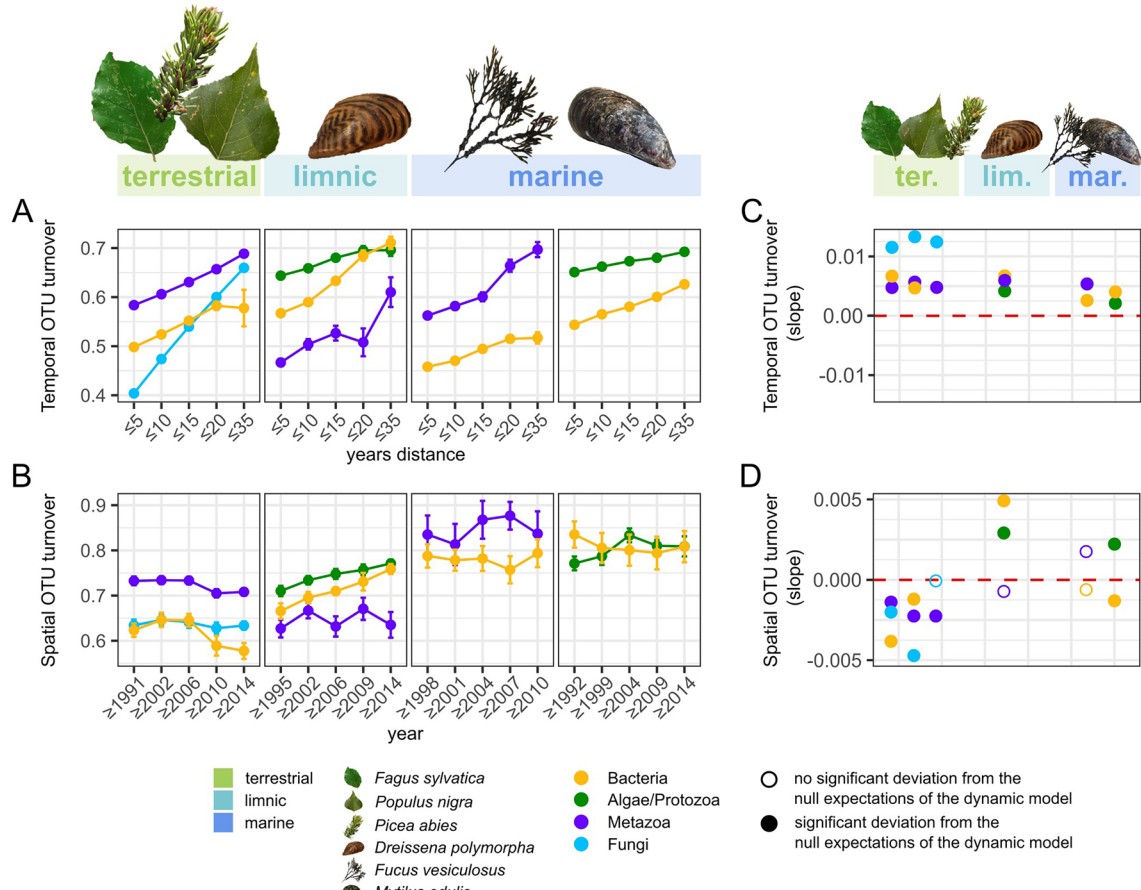

**Extended Data Fig. 5 | Multidecadal trends of temporal and spatial OTU turnover in communities across the tree of life associated with natural DNA sampler organisms from the ESB. a**) Temporal changes in community composition (β-diversity measured using turnover component of Jaccard distance) as a function of the time interval (in years) between samples from the same sampling site. **b**) Trends in spatial β-diversity (degree of dissimilarity in community composition between different sampling locations, measured using turnover component of Jaccard distance) of the associated communities

from 1991 to 2021. All diversity indices were summarized as mean with standard error bars across sampling locations and/or time windows (and for terrestrial samples across tree species; for underlying sample sizes see data availability statement). **c**) + **d**) Diversity trends from **a**) + **b**) reduced to their respective slopes. Filled circles indicate significant departures from the null expectations generated through the dynamic model for community assembly, suggesting an out-of-equilibrium dynamic. Different ecosystems and taxonomic groups are represented by different colors. Icons refer to sampled species.

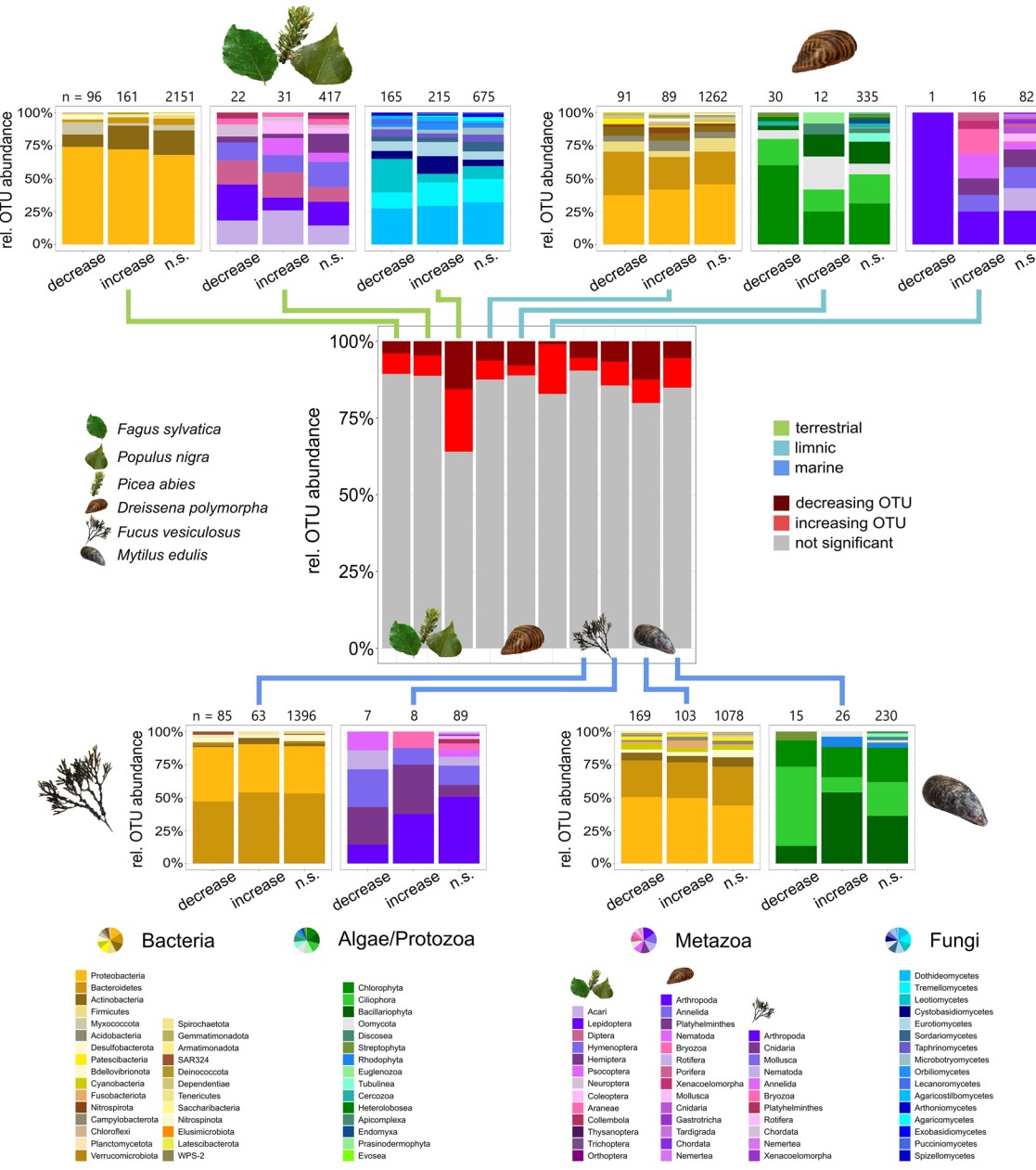

**Extended Data Fig. 6 | Proportion of OTUs showing a significant increase, decrease, or no significant trend in occurrence over time and the respective taxonomic composition of each category (increase, decrease, n.s.) for each sampling type and taxonomic group (Bacteria, Algae/Protozoa, Metazoa, Fungi).** The central chart represents the proportion of OTUs increasing, decreasing or showing no significant trend per taxonomic group and sample type. From each bar of the central chart a line (indicating the ecosystem type by its color) leads to one of the surrounding bar charts, which illustrates the taxonomic composition of each trend category (increase, decrease, n.s.). Community composition is shown on phylum level for bacteria, microeukaryotic plankton, and aquatic metazoans. Terrestrial metazoans are shown on order level and fungi on class level. Different ecosystems, trend categories and taxonomic groups are represented by different color palettes. Icons refer to sampled species.

**Extended Data Table 1 | Estimates of regional diversity indicate that the sampling effort is sufficient to capture most of the diversity**

| Dataset | Number of observed OTUs | Estimated regional diversity |
|---|---|---|
| Beech (16S) | 3,469 | 3,572 |
| Beech (COI) | 992 | 1,038 |
| Beech (ITS) | 1,773 | 1,829 |
| Bladderwrack (16S) | 3,865 | 3,976 |
| Bladderwrack (18S) | 399 | 417 |
| Blue mussel (16S) | 6,724 | 7,007 |
| Blue mussel (18S) | 2,207 | 2,311 |
| Poplar (16S) | 1,973 | 2,040 |
| Poplar (COI) | 395 | 412 |
| Poplar (ITS) | 761 | 788 |
| Spruce (COI) | 1,064 | 1,105 |
| Spruce (ITS) | 1,901 | 1,962 |
| Zebra mussel (16S) | 5,891 | 6,153 |
| Zebra mussel (18S Algae) | 3,043 | 3,183 |
| Zebra mussel (18S Metazoa) | 320 | 332 |

For each dataset, we reported the observed number of OTUs and the regional diversity estimated using bootstrapping (ACE estimates of γ-diversity).

**Extended Data Table 2 | Validation of our approach using simulations**

Simulated scenario 1: No disturbance

|  | Temporal trend in OTU richness | Temporal β-diversity | Temporal trend in spatial β-diversity | Temporal trends in γ-gamma diversity |
|---|---|---|---|---|
| Significant | 0 | 10 | 2 | 0 |
| Non-significant | 10 | 0 | 8 | 10 |

Simulated scenario 2: Regional diversity decline and spatial homogenization

|  | Temporal trend in OTU richness | Temporal β-diversity | Temporal trend in spatial β-diversity | Temporal trends in γ-gamma diversity |
|---|---|---|---|---|
| Significant | 0 | 10 | 10 | 10 |
| Non-significant | 10 | 0 | 0 | 0 |

For each simulated scenario, we reported the number of simulations with significant or non-significant temporal diversity trends evaluated by comparing them to null expectations under a non-neutral dynamic model for community assembly derived from the equilibrium theory of island biogeography (ETIB). Blue backgrounds indicate the results that are expected for each simulated scenario.

**Extended Data Table 3 | Significance of the observed temporal diversity trends evaluated by comparing them to null expectations under a non-neutral dynamic model for community assembly derived from the equilibrium theory of island biogeography (ETIB)**

| Dataset | Temporal trend in α-diversity | Annual temporal β-diversity (mean β-diversity changes in 1-year intervals) | Temporal trend in spatial β-diversity | Temporal trend in γ-diversity |
|---|---|---|---|---|
| Beech (16S) | slope=-1.9, p=0.002 | mean=0.59, p<0.001 | slope=-0.0021, p<0.001 | slope=-16.8, p<0.001 |
| Beech (COI) | slope=-0.44, p=0.055 | mean=0.63, p<0.001 | slope=-0.0015, p=0.005 | slope=-0.4, p=0.268 |
| Beech (ITS) | slope=-0.42, p=0.137 | mean=0.47, p<0.001 | slope=-7e-04, p=0.055 | slope=-1.22, p=0.154 |
| Bladderwrack (16S) | slope=-4.95, p=0.001 | mean=0.48, p<0.001 | slope=0, p=0.489 | slope=-14.37, p<0.001 |
| Bladderwrack (18S) | slope=0.52, p=0.091 | mean=0.63, p<0.001 | slope=0.0023, p=0.077 | slope=1.67, p=0.051 |
| Blue mussel (16S) | slope=-5.64, p<0.001 | mean=0.58, p<0.001 | slope=-0.0011, p<0.001 | slope=-19.91, p<0.001 |
| Blue mussel (18S) | slope=2.04, p<0.001 | mean=0.69, p<0.001 | slope=-3e-04, p=0.168 | slope=5.86, p<0.001 |
| Poplar (16S) | slope=2.3, p<0.001 | mean=0.57, p<0.001 | slope=-0.0035, p<0.001 | slope=-0.86, p=0.232 |
| Poplar (COI) | slope=0.4, p=0.021 | mean=0.68, p<0.001 | slope=-0.0012, p=0.125 | slope=0.62, p=0.056 |
| Poplar (ITS) | slope=2.08, p<0.001 | mean=0.46, p<0.001 | slope=-0.0035, p<0.001 | slope=2.17, p=0.003 |
| Spruce (COI) | slope=1.12, p<0.001 | mean=0.58, p<0.001 | slope=-0.0024, p<0.001 | slope=0.34, p=0.242 |
| Spruce (ITS) | slope=1.51, p<0.001 | mean=0.38, p<0.001 | slope=1e-04, p=0.303 | slope=5.7, p<0.001 |
| Zebra mussel (16S) | slope=4.06, p<0.001 | mean=0.6, p<0.001 | slope=0.0034, p<0.001 | slope=46.4, p<0.001 |
| Zebra mussel (18S Algae) | slope=-1.57, p<0.001 | mean=0.68, p<0.001 | slope=0.0027, p<0.001 | slope=3.75, p=0.644 |
| Zebra mussel (18S Metazoa) | slope=0.65, p<0.001 | mean=0.56, p<0.001 | slope=-5e-04, p=0.354 | slope=2.99, p=0.002 |

This table indicates the temporal trends in α-diversity (OTU richness through time; first column), the annual temporal β-diversity (mean β-diversity changes within one community in one year; second column), the temporal trends in spatial β-diversity (β-diversity across different sites through time; third column), and the temporal trends in γ-diversity (last column). Measures of β-diversity are computed using Jaccard distances. The significance of the empirical temporal trends was assessed by comparing them with the simulated trends. An empirical trend was considered significant if it fell outside the central 95% of the simulated distribution; p-values were computed as the proportions of simulated trends that were higher or lower than the empirical trend (see Methods). Green backgrounds indicate significant diversity increases toward the present, whereas purple backgrounds stand for significant diversity decreases. Gray backgrounds represent annual temporal β-diversity significantly larger than expected.

# Reporting Summary

## Statistics

For all statistical analyses, confirm that the following items are present in the figure legend, table legend, main text, or Methods section.

| n/a | Confirmed | |
|---|---|---|
| ☐ | ☒ | The exact sample size (*n*) for each experimental group/condition, given as a discrete number and unit of measurement |
| ☐ | ☒ | A statement on whether measurements were taken from distinct samples or whether the same sample was measured repeatedly |
| ☐ | ☒ | The statistical test(s) used AND whether they are one- or two-sided *Only common tests should be described solely by name; describe more complex techniques in the Methods section.* |
| ☒ | ☐ | A description of all covariates tested |
| ☐ | ☒ | A description of any assumptions or corrections, such as tests of normality and adjustment for multiple comparisons |
| ☐ | ☒ | A full description of the statistical parameters including central tendency (e.g. means) or other basic estimates (e.g. regression coefficient) AND variation (e.g. standard deviation) or associated estimates of uncertainty (e.g. confidence intervals) |
| ☐ | ☒ | For null hypothesis testing, the test statistic (e.g. $F$, $t$, $r$) with confidence intervals, effect sizes, degrees of freedom and $P$ value noted *Give P values as exact values whenever suitable.* |
| ☒ | ☐ | For Bayesian analysis, information on the choice of priors and Markov chain Monte Carlo settings |
| ☒ | ☐ | For hierarchical and complex designs, identification of the appropriate level for tests and full reporting of outcomes |
| ☒ | ☐ | Estimates of effect sizes (e.g. Cohen's *d*, Pearson's *r*), indicating how they were calculated |

*Our web collection on statistics for biologists contains articles on many of the points above.*

## Software and code

Policy information about availability of computer code

| Data collection | All data analyzed in this study were generated for this purpose (see next section). |
|---|---|
| Data analysis | bioinformatic scripts. For data analyses and statistics, we used R 4.3.1. with the packages scales_1.3.0, phytools_2.3-0, maps_3.4.2, ape_5.8, betapart_1.6, RColorBrewer_1.1-3, nlme_3.1-164, ggplot2_3.5.1, vegan_2.6-6.1, lattice_0.22-5 and permute_0.9-7. We also used python 3 with the libraries numpy, keras and sklearn for training the artificial neural networks. All the scripts we used and the custom algorithms we developed (included the non-neutral model for community ecology and the simulations) are freely available to editors and reviewers following this private link https://www.scidb.cn/en/s/U7NFv2; DOI: 10.57760/sciencedb.13553 (see Code availability section). |

For manuscripts utilizing custom algorithms or software that are central to the research but not yet described in published literature, software must be made available to editors and reviewers. We strongly encourage code deposition in a community repository (e.g. GitHub). See the Nature Portfolio guidelines for submitting code & software for further information.

## Data

Policy information about availability of data

All manuscripts must include a data availability statement. This statement should provide the following information, where applicable:

- Accession codes, unique identifiers, or web links for publicly available datasets
- A description of any restrictions on data availability
- For clinical datasets or third party data, please ensure that the statement adheres to our policy

> Raw Illumina sequencing data will be available in the ENA repository under the accession number PRJEB88877.
> OTU tables, supplementary data, source data files for figures, and sample sizes underlying all box plots and means with error bars shown are available in the Science Data Bank (DOI: 10.57760/sciencedb.13553). Private link for editors and reviewers: https://www.scidb.cn/en/s/U7NFv2

## Research involving human participants, their data, or biological material

Policy information about studies with human participants or human data. See also policy information about sex, gender (identity/presentation), and sexual orientation and race, ethnicity and racism.

| | |
|---|---|
| Reporting on sex and gender | - |
| Reporting on race, ethnicity, or other socially relevant groupings | - |
| Population characteristics | - |
| Recruitment | - |
| Ethics oversight | - |

Note that full information on the approval of the study protocol must also be provided in the manuscript.

# Field-specific reporting

Please select the one below that is the best fit for your research. If you are not sure, read the appropriate sections before making your selection.

☐ Life sciences          ☐ Behavioural & social sciences          ☒ Ecological, evolutionary & environmental sciences

For a reference copy of the document with all sections, see nature.com/documents/nr-reporting-summary-flat.pdf

# Ecological, evolutionary & environmental sciences study design

All studies must disclose on these points even when the disclosure is negative.

| | |
|---|---|
| Study description | We used 38 years of archived samples of the German Environmental Specimen Bank, namely tree leaves (Fagus sylvatica, Picea abies and Populus nigra), mussel tissue (Dreissena polymorpha and Mytilus edulis) and coastal macroalgae (Fucus vesiculosus), as natural eDNA samplers to analyze retrospective biodiversity trends using a metabarcoding approach.<br>We looked at alpha as well as temporal, spatial beta and gamma-diversity patterns across the whole of Germany and compared them to the null-expectations of a non-neutral model for community ecology.<br>We used two to three primer pairs for each sampling species and performed three to six PCR replicates per sample to analyze metazoan, algal/protozoan, fungal and bacterial communities. |
| Research sample | We analyzed tree leaves (Fagus sylvatica, Picea abies and Populus nigra), mussel tissue (Dreissena polymorpha and Mytilus edulis) and coastal macroalgae (Fucus vesiculosus) since these species are routinely sampled by the German Environmental Specimen Bank (GESB) according to highly standardized sampling guidelines, which are publicly available on the homepage of the GESB. Samples have been perfectly preserved over liquid nitrogen since the time of collection. |
| Sampling strategy | Samples have been collected by the German Environmental Specimen Bank according to highly standardizes guidelines. No sample-size calculation was performed. All available samples were used for the analysis to achieve a time series as long and as complete as possible. |
| Data collection | Metabarcoding data was generated from Illumina sequencing at Trier University. |
| Timing and spatial scale | Earliest samples were collected in 1985 and latest in 2022. Sampling was performed annually or biannually. |
| Data exclusions | Some samples were excluded from the statistical analysis to achieve consistent time series. For details see Methods section "Statistical model and analyses of community diversity". |

| Reproducibility | The German Environmental Specimen Bank (GESB) uses highly standardized and publicly available sampling guidelines to secure sampling reproducibility. These guidelines can be found on the GESB homepage. For laboratory work we used established protocols for DNA isolation, DNA amplification and Illumina sequencing. For details see Methods section "DNA extraction, library preparation, sequencing and sequence processing" and supplementary Table 5 "Details of the laboratory workflow". |
|---|---|
| Randomization | Samples were allocated into groups according to sampling species, since they represent different ecosystems (marine, limnic and terrestrial). Tree species were allocated into one group, since they represent the terrestrial ecosystem. Additionally, trends for the terrestrial ecosystem are shown in Extended Data Figure 3 to provide detailed information for each tree species. |
| Blinding | Blinding was not relevant for this study since all data sets were treated equally. |

Did the study involve field work?  ☒ Yes  ☐ No

# Field work, collection and transport

| Field conditions | Required field conditions depend on sampling species. Detailed information for each sampling species is listed in the respective GESB guideline. |
|---|---|
| Location | Species Location Latitude_WGS84 Longitude_WGS84
European Beech (Fagus sylvatica) Bayerischer Wald 48.966395 13.430375
European Beech (Fagus sylvatica) Belauer See 54.06061 10.15373
European Beech (Fagus sylvatica) Berchtesgaden 47.56574 12.89274
European Beech (Fagus sylvatica) Harz 51.838522 10.635239
European Beech (Fagus sylvatica) Pfälzerwald 49.145482 7.713357
European Beech (Fagus sylvatica) Scheyern 48.487405 11.428479
European Beech (Fagus sylvatica) Solling 51.73638 9.57365
Bladderwrack (Fucus vesiculosus) Eckwarderhörne 53.519772 8.231447
Bladderwrack (Fucus vesiculosus) Ostsee 54.506479 13.284812
Bladderwrack (Fucus vesiculosus) Sylt 55.011 8.4125
Blue Mussel (Mytilus edulis) Eckwarderhörne 53.519772 8.231447
Blue Mussel (Mytilus edulis) Ostsee 54.275622 12.318046
Blue Mussel (Mytilus edulis) Sylt 55.011 8.4125
Lombardy Poplar (Populus nigra) Leipzig 51.353013 12.404477
Lombardy Poplar (Populus nigra) Saartal 49.22591 7.00576
Norway Spruce (Picea abies) Bayerischer Wald 48.966921 13.435102
Norway Spruce (Picea abies) Belauer See 54.10381 10.24531
Norway Spruce (Picea abies) Berchtesgaden 47.56113 12.89025
Norway Spruce (Picea abies) Harz 51.792691 10.645409
Norway Spruce (Picea abies) Pfälzerwald 49.0902 7.43575
Norway Spruce (Picea abies) Scheyern 48.487405 11.428479
Norway Spruce (Picea abies) Solling 51.787381 9.610741
Zebramussel (Dreissena polymorpha) Bimmen 51.858614 6.073054
Zebramussel (Dreissena polymorpha) Blankenese 53.556886 9.80897
Zebramussel (Dreissena polymorpha) Cumlosen 53.03962009 11.63766977
Zebramussel (Dreissena polymorpha) Jochenstein 48.566659 13.60556
Zebramussel (Dreissena polymorpha) Koblenz 50.34718 7.60211
Zebramussel (Dreissena polymorpha) Prossen 50.92708 14.11624
Zebramussel (Dreissena polymorpha) Rehlingen 49.371943 6.69865
Zebramussel (Dreissena polymorpha) Ulm 48.33667 9.93357
Zebramussel (Dreissena polymorpha) Zehren 51.26445 13.4026 |
| Access & import/export | Habitat access and permits are managed by the German Environmental Specimen Bank. |
| Disturbance | No disturbance was noticed. |

# Reporting for specific materials, systems and methods

We require information from authors about some types of materials, experimental systems and methods used in many studies. Here, indicate whether each material, system or method listed is relevant to your study. If you are not sure if a list item applies to your research, read the appropriate section before selecting a response.

## Materials & experimental systems

| n/a | Involved in the study |
|---|---|
| ☒ | ☐ Antibodies |
| ☒ | ☐ Eukaryotic cell lines |
| ☒ | ☐ Palaeontology and archaeology |
| ☐ | ☒ Animals and other organisms |
| ☒ | ☐ Clinical data |
| ☒ | ☐ Dual use research of concern |
| ☐ | ☒ Plants |

## Methods

| n/a | Involved in the study |
|---|---|
| ☒ | ☐ ChIP-seq |
| ☒ | ☐ Flow cytometry |
| ☒ | ☐ MRI-based neuroimaging |

# Animals and other research organisms

Policy information about studies involving animals; ARRIVE guidelines recommended for reporting animal research, and Sex and Gender in Research

| Laboratory animals | The study did not involve laboratory animals. |
|---|---|
| Wild animals | Dreissena polymorpha and Mytilus edulis were collected and stored above liquid nitrogen in the field. The GESB dissects frozen soft tissue including respiratory water and separates it from the shell. The soft tissue incl. respiratory water is ground to a fine powder. Samples are always stored below -130°C. For details see methods section in the manuscript. |
| Reporting on sex | No information on the sex is recorded. |
| Field-collected samples | Dreissena polymorpha and Mytilus edulis were collected in the field and immediately stored above liquid nitrogen. |
| Ethics oversight | No ethical approval or guidance was needed since the samples were collected by the German Environmental Specimen Bank which is authorized by the Central Environmental Agency of Germany. |

Note that full information on the approval of the study protocol must also be provided in the manuscript.

# Plants

| Seed stocks | For detailed information of collection location see part "Methods - Specimen Bank Data". Plant specimen were collected according to highly standardized guidelines by the German Environmental Specimen Bank which can be found on the homepage. |
|---|---|
| Novel plant genotypes | - |
| Authentication | - |

