## [Peer Review File · Nature Ecology & Evolution]

Archived natural DNA samplers reveal four decades of biodiversity change across the tree of life

Corresponding Author: Dr Henrik Krehenwinkel

This manuscript has been previously reviewed at another journal. This document only contains information relating to versions considered at Nature Ecology & Evolution.

Version 0:

Reviewer comments:

Reviewer #1

(Remarks to the Author)

In all, this appears to be a very interesting and potentially quite useful manuscript. The main positive aspects are 1) the data sources, 2) the modelling approach based on ETIB. In all, I think this will be a good contribution. That said, I cannot immediately recommend for publication, nor can I fully evaluate the results, because of the way beta-diversity was measured. Although implied that temporal and beta diversity are being measured, from the methods, it appears that only one small part of beta diversity—the turnover component—is being measured. It's absolutely a useful part of beta-diversity, but only one part. The way the manuscript is written, and the questions asked/addressed do not match this particular kind of beta diversity. I would much prefer to see a more complete exploration of beta diversity and how it's changing. The currently implication of 'biotic homogenization' is not entirely correct or consistent with the literature, even papers cited.

I particularly emphasize the paper by Blowes et al 2024 Science advances, where a more complete exposition of beta diversity and homogenization is offered. I also note that while the abstract and much of the discussion is on 'homogenization', even with the narrow definition used here, this did not seem to always (or even often) be the case. If the authors really want to discuss homogenization, they need to consider the framework of Blowes et al. Declining beta diversity need not imply declining gamma diversity. Especially with a narrow definition of beta diversity. Again, I think this is a good study. However, I cannot evaluate the results with the current metrics, and would encourage the authors to consider a more complete exposition of beta diversity, which continues to be confused and confusing in the literature. Both for time and for space.

(Remarks on code availability)

Reviewer #2

(Remarks to the Author)

Here, Junk et al. investigate shifts in diversity across the tree of life derived from amazingly well kept and preserved 'natural samplers' of biodiversity, such as forest tree leaves, mussels, marine macroalgae across multi-decadal scales in Germany. They found biotic homogenisation in forest canopy data, but not in aquatic systems, but general trends in turnover across the board. Regarding the terrestrial canopy findings, it mirrors (or are subsamples of?) the arthropod data from Krehenwinkel et al. (2022 eLife), but I do not feel that there is enough reference to that paper explicitly in this one. Without a doubt, the present manuscript stands on its own, but I feel that more cross talk between the two would be useful. For example, what if the biotic homogenisation that we see in Fig. 2F is actually being driven by the host microbiomes of the arthropods that we know to have shown homogenisation in the 2022 paper? Could this be tested via network/co-occurrence, JDSM or similar approaches?

The datasets and vision are amazing and the presentation of the analyses are high quality, but there are areas of the writing/referencing which, I feel, would benefit from some fresh eyes and doubling down on attention to detail and referencing.

There are also question marks re: the modelling, in that, is the model new to this analysis and are there any guarantees that the hypotheses being tested with the model are yielding plausible results. What confidence can the team assert re: the modelling aspect?

Abstract

The front end of the manuscript has an immature writing feel about it. The first few sentences of the Abstract/whole Abstract do not read well and need attention to flow and ease of reading.

Line 28 – Only mentions terrestrial ecosystems, but doesn't the work focus on multiple biomes? What about the freshwater and marine findings?

Line 29 – cf. biodiversity loss. I understand the meaning, but this reads as a contradiction to the previous few lines (no localized declines). Does the paper not report turnover, not massive loss?

Main

The introduction felt rushed; the first sentence has no detail/references. I felt that the underpinning scientific rationale for the study was absent. The introduction was quite odd?

The first mention of ESBs is not referenced – how many other countries adhere to these sort of cryogenically preserved ESBs?

Line 42 – Sentence starts with abbreviation.

Line 46 – metaorganism – is this the right term (no reference)? Online searches suggests metaorganisms are host and microbiome, which is not the concept being exploited here? Consequently, I do not think sentence 44, or the following sentence makes any sense?

Line 58 – how do they know they are characterizing canopy-associated fungi and bacteria and not just an aggregation of all?

Hypotheses – It feels like H2 contradicts H1?

Is it predicted that fungal diversity will be higher than bacterial diversity in the tree leaf sample data?

Figure 1A is not introduced in the text?

Line 88 – how do these samples/analyses differ from the Krehenwinkel et al. eLife paper? Is this dataset completely novel, or are some data shared?

Lines 94-95 – Is it likely that bacterial diversity is ca. 5K OTUs, whereas metazoan richness is 3.2K OTUs; it seems intuitive that either the bacteria are low and the metazoans are high. Do these data add up to the team – what do contemporary richness estimates look like from forest ecosystems. If this represents an unexpected skew, what explanations/tests could the team do/suggest?

Line 115 – Baltic, North, salinity data – no references?

Lines 142-143 – Is the model being used here novel to this paper, or has been published elsewhere? I could not see any of the team in the references cited? If the model is new, does it need to be benchmarked on sample/existing data to check the assumptions/parameters are working as expected?

Line 151 – strongly – compared to what?

Line 155 – individual?

Line 180 – changing env.conditions – which ones, based on what data?

Line 203 – Danube, Rhine biogeography – no reference cited?

Line 227 – stability and functional redundancy – is there any evidence that the ecosystems are unstable?

Line 391 – All these taxa is really vague – can you provide more information?

Line 415 – check journal style wrt numbers and in text written citations.

Line 416 – under(neath?) clean benches – do they mean on clean benches?

Line 418 – 200mg – does reference 9 only refer to the leaf samples; if so, this needs to be clear.

Line 421 – barcode – metabarcode?

Line 462 – Wed excluded?

Line 466 – checked for contamination using negative controls – how? What data (or none?) were in the negatives? Line 478 – similar question. Where did the 0.01% threshold come from – the negatives, or random number?

Lines 481-487 – how many samples were involved in these comparisons?

Line 563 – year 1500 – why?

(Remarks on code availability)

NA

Decision Letter:

16th December 2024

Dear Dr Krehenwinkel,

Your Article, "Archived natural DNA samplers reveal four decades of biodiversity change across the tree of life" has now been seen by 2 reviewers. You will see from their comments copied below that while they find your work of considerable potential interest, they have raised quite substantial concerns that must be addressed. In light of these comments, we cannot accept the manuscript for publication, but would be very interested in considering a revised version that addresses these serious concerns.

We hope you will find the reviewers' comments useful as you decide how to proceed. If you wish to submit a substantially revised manuscript, please bear in mind that we will be reluctant to approach the reviewers again in the absence of major revisions. In particular, we are concerned by the comments from reviewer #1 and we would like to see a more thorough examination of beta diversity as suggested by this reviewer if we are to further consider the manuscript, as well as more support for your conclusions based on biotic homogenisation.

If you choose to revise your manuscript taking into account all reviewer and editor comments, please highlight all changes in the manuscript text file in Microsoft Word format.

* Include a "Response to reviewers" document detailing, point-by-point, how you addressed each referee comment. If no action was taken to address a point, you must provide a compelling argument. This response will be sent back to the referees along with the revised manuscript.

* If you have not done so already we suggest that you begin to revise your manuscript so that it conforms to our Article format instructions at <http://www.nature.com/natecolevol/info/final-submission>. Refer also to any guidelines provided in this letter.

Link Redacted

If you wish to submit a suitably revised manuscript we would hope to receive it within 6 months. If you cannot send it within this time, please let us know. We will be happy to consider your revision so long as nothing similar has been accepted for publication at Nature Ecology & Evolution or published elsewhere.

Nature Ecology & Evolution is committed to improving transparency in authorship. As part of our efforts in this direction, we are now requesting that all authors identified as 'corresponding author' on published papers create and link their Open Researcher and Contributor Identifier (ORCID) with their account on the Manuscript Tracking System (MTS), prior to acceptance. This applies to primary research papers only. ORCID helps the scientific community achieve unambiguous attribution of all scholarly contributions. You can create and link your ORCID from the home page of the MTS by clicking on 'Modify my Springer Nature account'. For more information please visit <http://www.springernature.com/orcid>.

Thank you for the opportunity to review your work.

[redacted]

Reviewer expertise:

Reviewer #1: ecology, biodiversity

Reviewer #2: molecular ecology, metabarcoding

Reviewers' comments:

Reviewer #1 (Remarks to the Author):

In all, this appears to be a very interesting and potentially quite useful manuscript. The main positive aspects are 1) the data sources, 2) the modelling approach based on ETIB. In all, I think this will be a good contribution. That said, I cannot immediately recommend for publication, nor can I fully evaluate the results, because of the way beta-diversity was measured. Although implied that temporal and beta diversity are being measured, from the methods, it appears that only one small part of beta diversity—the turnover component—is being measured. It's absolutely a useful part of beta-diversity, but only one part. The way the manuscript is written, and the questions asked/addressed do not match this particular kind of beta diversity. I would much prefer to see a more complete exploration of beta diversity and how it's changing. The currently implication of 'biotic homogenization' is not entirely correct or consistent with the literature, even papers cited.

I particularly emphasize the paper by Blowes et al 2024 Science advances, where a more complete exposition of beta diversity and homogenization is offered. I also note that while the abstract and much of the discussion is on 'homogenization', even with the narrow definition used here, this did not seem to always (or even often) be the case. If the authors really want to discuss homogenization, they need to consider the framework of Blowes et al. Declining beta diversity need not imply declining gamma diversity. Especially with a narrow definition of beta diversity. Again, I think this is a good study. However, I cannot evaluate the results with the current metrics, and would encourage the authors to consider a more complete exposition of beta diversity, which

continues to be confused and confusing in the literature. Both for time and for space.

Reviewer #2 (Remarks to the Author):

Here, Junk et al. investigate shifts in diversity across the tree of life derived from amazingly well kept and preserved 'natural samplers' of biodiversity, such as forest tree leaves, mussels, marine macroalgae across multi-decadal scales in Germany. They found biotic homogenisation in forest canopy data, but not in aquatic systems, but general trends in turnover across the board. Regarding the terrestrial canopy findings, it mirrors (or are subsamples of?) the arthropod data from Krehenwinkel et al. (2022 eLife), but I do not feel that there is enough reference to that paper explicitly in this one. Without a doubt, the present manuscript stands on its own, but I feel that more cross talk between the two would be useful. For example, what if the biotic homogenisation that we see in Fig. 2F is actually being driven by the host microbiomes of the arthropods that we know to have shown homogenisation in the 2022 paper? Could this be tested via network/co-occurrence, JDSM or similar approaches?

The datasets and vision are amazing and the presentation of the analyses are high quality, but there are areas of the writing/referencing which, I feel, would benefit from some fresh eyes and doubling down on attention to detail and referencing.

There are also question marks re: the modelling, in that, is the model new to this analysis and are there any guarantees that the hypotheses being tested with the model are yielding plausible results. What confidence can the team assert re: the modelling aspect?

Abstract

The front end of the manuscript has an immature writing feel about it. The first few sentences of the Abstract/whole Abstract do not read well and need attention to flow and ease of reading.

Line 28 – Only mentions terrestrial ecosystems, but doesn't the work focus on multiple biomes? What about the freshwater and marine findings?

Line 29 – cf. biodiversity loss. I understand the meaning, but this reads as a contradiction to the previous few lines (no localized declines). Does the paper not report turnover, not massive loss?

Main

The introduction felt rushed; the first sentence has no detail/references. I felt that the underpinning scientific rationale for the study was absent. The introduction was quite odd?

The first mention of ESBs is not referenced – how many other countries adhere to these sort of cryogenically preserved ESBs? Line 42 – Sentence starts with abbreviation.

Line 46 – metaorganism – is this the right term (no reference)? Online searches suggests metaorganisms are host and microbiome, which is not the concept being exploited here? Consequently, I do not think sentence 44, or the following sentence makes any sense?

Line 58 – how do they know they are characterizing canopy-associated fungi and bacteria and not just an aggregation of all?

Hypotheses – It feels like H2 contradicts H1?

Is it predicted that fungal diversity will be higher than bacterial diversity in the tree leaf sample data?

Figure 1A is not introduced in the text?

Line 88 – how do these samples/analyses differ from the Krehenwinkel et al. eLife paper? Is this dataset completely novel, or are some data shared?

Lines 94-95 – Is it likely that bacterial diversity is ca. 5K OTUs, whereas metazoan richness is 3.2K OTUs; it seems intuitive that either the bacteria are low and the metazoans are high. Do these data add up to the team – what do contemporary richness estimates look like from forest ecosystems. If this represents an unexpected skew, what explanations/tests could the team do/suggest?

Line 115 – Baltic, North, salinity data – no references?

Lines 142-143 – Is the model being used here novel to this paper, or has been published elsewhere? I could not see any of the team in the references cited? If the model is new, does it need to be benchmarked on sample/existing data to check the assumptions/parameters are working as expected?

Line 151 – strongly – compared to what?

Line 155 – individual?

Line 180 – changing env.conditions – which ones, based on what data?

Line 203 – Danube, Rhine biogeography – no reference cited?

Line 227 – stability and functional redundancy – is there any evidence that the ecosystems are unstable?

Line 391 – All these taxa is really vague – can you provide more information?

Line 415 – check journal style wrt numbers and in text written citations.

Line 416 – under(neath?) clean benches – do they mean on clean benches?

Line 418 – 200mg – does reference 9 only refer to the leaf samples; if so, this needs to be clear.

Line 421 – barcode – metabarcode?

Line 462 – Wed excluded?

Line 466 – checked for contamination using negative controls – how? What data (or none?) were in the negatives? Line 478 – similar question. Where did the 0.01% threshold come from – the negatives, or random number?

Lines 481-487 – how many samples were involved in these comparisons?

Line 563 – year 1500 – why?

Reviewer #2 (Remarks on code availability):

NA

Version 1:

Reviewer comments:

Reviewer #1

(Remarks to the Author)

I would like to thank the authors for taking my review seriously and providing a more comprehensive look at the beta diversity problem. It's great that the results are similar, but I think the newer analyses enhance the work and connections to other work. I have nothing further.

(Remarks on code availability)

Reviewer #2

(Remarks to the Author)

I can see that the authors have attended to the breadth of my and R1's comments and judging from the track changed version of the manuscript, they have performed comprehensive edits to improve flow, accuracy, and message. I cannot fully understand why the previous version was submitted, but I am happy with the revisions and I hope that the authors will share this vision. I hope that the manuscript will move to the publication process swiftly and be well cited in the future.

(Remarks on code availability)

NA

Decision Letter:

27th March 2025

Dear Dr. Krehenwinkel,

Thank you for submitting your revised manuscript "Archived natural DNA samplers reveal four decades of biodiversity change across the tree of life" (NATECOLEVOL-24102798A). It has now been seen again by the original reviewers and their comments are below. The reviewers find that the paper has improved in revision, and therefore we'll be happy in principle to publish it in Nature Ecology & Evolution, pending minor revisions to satisfy the reviewers' final requests and to comply with our editorial and formatting guidelines.

If you have not done so already, please ensure that you also email us completed copies of the Reporting summary and Editorial policy checklists:

Reporting summary: <https://www.nature.com/documents/nr-reporting-summary.pdf>

Editorial policy checklist: <https://www.nature.com/documents/nr-editorial-policy-checklist.pdf>

[redacted]

Reviewer #1 (Remarks to the Author):

I would like to thank the authors for taking my review seriously and providing a more comprehensive look at the beta diversity problem. It's great that the results are similar, but I think the newer analyses enhance the work and connections to other work. I have nothing further.

Reviewer #2 (Remarks to the Author):

I can see that the authors have attended to the breadth of my and R1's comments and judging from the track changed version of

the manuscript, they have performed comprehensive edits to improve flow, accuracy, and message. I cannot fully understand why the previous version was submitted, but I am happy with the revisions and I hope that the authors will share this vision. I hope that the manuscript will move to the publication process swiftly and be well cited in the future.

Reviewer #2 (Remarks on code availability):

NA

Version 2:

Decision Letter:

25th June 2025

Dear Dr Krehenwinkel,

We are pleased to inform you that your Article entitled "Archived natural DNA samplers reveal four decades of biodiversity change across the tree of life", has now been accepted for publication in Nature Ecology & Evolution.

Over the next few weeks, your paper will be copyedited to ensure that it conforms to Nature Ecology and Evolution style. Once your paper is typeset, you will receive an email with a link to choose the appropriate publishing options for your paper and our Author Services team will be in touch regarding any additional information that may be required

Due to the importance of these deadlines, we ask you please us know now whether you will be difficult to contact over the next month. If this is the case, we ask you provide us with the contact information (email, phone and fax) of someone who will be able to check the proofs on your behalf, and who will be available to address any last-minute problems . Once your paper has been scheduled for online publication, the Nature press office will be in touch to confirm the details.

Acceptance of your manuscript is conditional on all authors' agreement with our publication policies (see www.nature.com/authors/policies/index.html). In particular your manuscript must not be published elsewhere and there must be no announcement of the work to any media outlet until the publication date (the day on which it is uploaded onto our web site).

Authors may need to take specific actions to achieve [compliance](https://www.springernature.com/gp/open-research/funding/policy-compliance-faqs) with funder and institutional open access mandates. If your research is supported by a funder that requires immediate open access (e.g. according to [Plan S principles](https://www.springernature.com/gp/open-research/plan-s-compliance)) then you should select the gold OA route, and we will direct you to the compliant route where possible. For authors selecting the subscription publication route, the journal's standard licensing terms will need to be accepted, including [self-archiving and license to publish](https://www.nature.com/nature-portfolio/editorial-policies/self-archiving-and-license-to-publish). Those licensing terms will supersede any other terms that the author or any third party may assert apply to any version of the manuscript.

We welcome the submission of potential cover material (including a short caption of around 40 words) related to your manuscript; suggestions should be sent to Nature Ecology & Evolution as electronic files (the image should be 300 dpi at 210 x 297 mm in either TIFF or JPEG format). Please note that such pictures should be selected more for their aesthetic appeal than for their scientific content, and that colour images work better than black and white or grayscale images. Please do not try to design a cover with the Nature Ecology & Evolution logo etc., and please do not submit composites of images related to your work. I am sure you will understand that we cannot make any promise as to whether any of your suggestions might be selected for the cover of the journal.

Link Redacted

[redacted]

P.S. Click on the following link if you would like to recommend Nature Ecology & Evolution to your librarian
<http://www.nature.com/subscriptions/recommend.html#forms>

** Visit the Springer Nature Editorial and Publishing website at http://editorial-jobs.springernature.com?utm_source=ejp_NEcoE_email&utm_medium=ejp_NEcoE_email&utm_campaign=ejp_NEcoE for more information about our career opportunities. If you have any questions please click [here](mailto:editorial.publishing.jobs@springernature.com).**

Reviewer #1 (Remarks to the Author):

In all, this appears to be a very interesting and potentially quite useful manuscript. The main positive aspects are 1) the data sources, 2) the modelling approach based on ETIB. In all, I think this will be a good contribution. That said, I cannot immediately recommend for publication, nor can I fully evaluate the results, because of the way beta-diversity was measured. Although implied that temporal and beta diversity are being measured, from the methods, it appears that only one small part of beta diversity—the turnover component—is being measured. It's absolutely a useful part of beta-diversity, but only one part. The way the manuscript is written, and the questions asked/addressed do not match this particular kind of beta diversity. I would much prefer to see a more complete exploration of beta diversity and how it's changing. The currently implication of 'biotic homogenization' is not entirely correct or consistent with the literature, even papers cited.

I particularly emphasize the paper by Blowes et al 2024 Science advances, where a more complete exposition of beta diversity and homogenization is offered. I also note that while the abstract and much of the discussion is on 'homogenization', even with the narrow definition used here, this did not seem to always (or even often) be the case. If the authors really want to discuss homogenization, they need to consider the framework of Blowes et al. Declining beta diversity need not imply declining gamma diversity. Especially with a narrow definition of beta diversity. Again, I think this is a good study. However, I cannot evaluate the results with the current metrics, and would encourage the authors to consider a more complete exposition of beta diversity, which continues to be confused and confusing in the literature. Both for time and for space.

- Thank you for your positive evaluation and your helpful criticism. We have revised our analyses and the manuscript accordingly.

First, we now calculated the full beta diversity (turnover + nestedness), reported these new results obtained with the full diversity in the main text (Figure 4) and moved the previous analyses made with the turnover component to the extended data (Extended Data Figure 4). The biodiversity patterns we report do not, however, differ between the two ways of computing beta diversity (lines 163-165, Figure 4, Extended Data Figure 4).

Second, our manuscript now also considers the conceptual framework of Blowes et al. (2024; Figure 5, lines 219-236). Looking at simultaneous variations of both alpha and gamma diversities, we now show that spatial homogenization is present in most of the studied ecosystems (in 9 out of 15 ecosystems; see Figure 5 and lines 220-222). Thanks to the framework of Blowes et al. 2024, we can now distinguish cases where the spatial homogenization is linked to a loss of gamma diversity (e.g. in bacterial communities of Beech, Blue mussels and Bladderwrack) and cases where the spatial homogenization is instead linked to the spread of invasive species across the different sites (increase of gamma diversity; e.g. in fungal communities of Poplars, lines 228-236).

Third, we reduced our discussion of homogenization to the terrestrial and marine samples, while emphasizing that the limnic communities showed the opposite trend (biotic differentiation). We also added concrete examples of species that may drive these patterns

(invasive species or extinct species; lines 222-226). We hope that this will make our results more clear.

Reviewer #2 (Remarks to the Author):

Here, Junk et al. investigate shifts in diversity across the tree of life derived from amazingly well kept and preserved 'natural samplers' of biodiversity, such as forest tree leaves, mussels, marine macroalgae across multi-decadal scales in Germany. They found biotic homogenisation in forest canopy data, but not in aquatic systems, but general trends in turnover across the board. Regarding the terrestrial canopy findings, it mirrors (or are subsamples of?) the arthropod data from Krehenwinkel et al. (2022 eLife), but I do not feel that there is enough reference to that paper explicitly in this one. Without a doubt, the present manuscript stands on its own, but I feel that more cross talk between the two would be useful.

- Thank you very much for the positive assessment of our work. We have now extended our discussion and better incorporated the Krehenwinkel et al. (2022) work on forest arthropods (lines 256-258). In fact, the data we presented here is almost entirely new with only little data shared. For the previous work on arthropods, we used a different DNA isolation protocol and only samples collected until 2018. We have now used a better suited protocol to deal with PCR inhibitors and hence isolated and processed most samples again. We also expanded the time series going from 1985 to 2022 (lines 353-358).

For example, what if the biotic homogenisation that we see in Fig. 2F is actually being driven by the host microbiomes of the arthropods that we know to have shown homogenisation in the 2022 paper? Could this be tested via network/co-occurrence, JDSM or similar approaches?

- The idea that the patterns of biotic homogenization found for forest arthropods could be mirrored by microbes is indeed a very good one. However, the DNA of arthropods is probably limited to very small traces in those leaf samples, while the amount of plant biomass and plant DNA vastly dominates the community. Hence, we consider it much more likely that the fungal and bacterial taxa we recover are plant derived and not associated with the insects in the sample. To explore this in more detail, we have functionally analyzed our microbial community data. We have identified all microbial lineages, which are typically arthropod associated, for example endosymbionts like *Wolbachia*, *Rickettsia*, *Cardinium* and *Buchnera*. Typical arthropod-associated bacteria make up only 4.2 % of the OTUs in the terrestrial community. The amount of arthropod-associated fungi is only 0.2 %. In contrast, of all OTUs we could functionally annotate on the genus level, typical plant associated microbes make up 34.2 % for fungi and 75.5 % for bacteria (lines 451-454). We thus consider it very unlikely that the pattern is driven by arthropod associated taxa. It is rather driven by true leaf associated microbes. Nevertheless, we have also repeated the analysis of diversity patterns after exclusion of typical arthropod associated microbial taxa and all the patterns remain the same (Figure below, Extended Data Figure 3, Figure 4E-H). We have added additional information

on this in the methods (lines 445-450). We have not included this figure in the main text, but could do so, if the reviewer deems this necessary.

Multidecadal trends of α -diversity, temporal and spatial β -diversity as well as γ -diversity in leaf-associated bacterial and fungal communities excluding arthropod-associated OTUs, with each tree species presented separately. A) Trends of OTU richness (α -diversity) of the associated

communities from 1991 to 2021. B) Temporal changes in community compositions (β -diversity measured using Jaccard distance) as a function of the time interval (in years) between samples from the same sampling site. C) Trends in spatial β -diversity (degree of dissimilarity in community composition between different sampling locations, measured using Jaccard distance) of the associated communities from 1991 to 2021. D) Bootstrap estimates of regional diversity (γ -diversity) for the associated communities from 1991 to 2021. All diversity indices were summarized as mean with standard error bars across sampling locations and/or time windows. E)-H) Diversity trends from A)-D) reduced to their respective slopes. Filled circles indicate significant departures from the null expectations through the dynamic model for community assembly, suggesting an out-of-equilibrium dynamic. Different taxonomic groups are represented by different colors. Icons refer to sampled species.

The datasets and vision are amazing and the presentation of the analyses are high quality, but there are areas of the writing/referencing which, I feel, would benefit from some fresh eyes and doubling down on attention to detail and referencing.

- Thanks again for your positive assessment. We have now reworked the entire manuscript and added additional citations throughout the text to better reflect the breadth of published work on the topic to date.

There are also question marks re: the modelling, in that, is the model new to this analysis and are there any guarantees that the hypotheses being tested with the model are yielding plausible results. What confidence can the team assert re: the modelling aspect?

- The model has been newly developed for our datasets particularly. We have now made it clearer in the main text (lines 69-72, 133-136, Figure 3). To test the validity of our modelling approach, we have now performed simulations and tested whether our approach correctly recovered the simulated trends, indicating that the hypotheses being tested with the model are yielding plausible results. We simulated two scenarios: 1) a scenario without any disturbance and 2) a scenario of biotic homogenization and regional diversity loss. In the first simulated scenario, we simply simulated community changes under the assumptions of our non-neutral dynamic model for community assembly and therefore did not expect any deviations from the null expectations when applying our approach. Conversely, in the second simulated scenario, we simulated the regional extinctions of 10% of the OTUs and their replacement by widespread invasive OTUs across all communities - we therefore expected no variation of α -diversities, a decrease of spatial β -diversities (spatial homogenization) as well as a decrease of γ -diversities (regional diversity loss). When applying our approach, we correctly recovered the simulated scenario in almost all the cases (Extended Data Table 2); confirming the validity of our approach. We have added descriptions of these simulations and

the test of the validity of the modelling approach in the methods section (lines 581-597 and Extended Data Table 2). We hence have full confidence that the model accurately detects the tested patterns.

Abstract

The front end of the manuscript has an immature writing feel about it. The first few sentences of the Abstract/whole Abstract do not read well and need attention to flow and ease of reading.

- We have rewritten the abstract and hope it is now better comprehensible.

Line 28 – Only mentions terrestrial ecosystems, but doesn't the work focus on multiple biomes? What about the freshwater and marine findings?

- The biotic homogenization is most evident in terrestrial ecosystems, but it can also be seen in some marine communities. We have tried to make this clearer in the current abstract and added our main findings for aquatic communities for completeness (lines 27-30).

Line 29 – cf. biodiversity loss. I understand the meaning, but this reads as a contradiction to the previous few lines (no localized declines). Does the paper not report turnover, not massive loss?

- This loss relates to biotic homogenization, e.g. a loss of biodiversity across space (at the regional scale). Furthermore, even while we find no local losses of species number at most individual sites, we have found a very clear turnover pattern. This turnover also involves the loss of species, with the only difference being that the loss is countered by the immigration of novel, possibly better adapted taxa. A replacement does not mean there is no loss. We rephrased the abstract and hope this is now better comprehensible.

Main

The introduction felt rushed; the first sentence has no detail/references. I felt that the underpinning scientific rationale for the study was absent. The introduction was quite odd? The first mention of ESBs is not referenced – how many other countries adhere to these sort of cryogenically preserved ESBs?

- Thank you for pointing this out. We have now given more space to the introduction with more attention to detail and references. We hope that this clarifies the scientific rationale for our study. We also added a reference for Environmental specimen banks (ESBs; line 42). In 2014, 28 ESBs have been established in countries across the globe (Zhao et al. 2014).

Line 42 – Sentence starts with abbreviation.

- We have introduced ESB as environmental specimen banks in the second paragraph of the introduction (lines 40-42). However, sentences starting with the term environmental specimen bank don't use the abbreviation anymore.

Line 46 – metaorganism – is this the right term (no reference)? Online searches suggests metaorganisms are host and microbiome, which is not the concept being exploited here? Consequently, I do not think sentence 44, or the following sentence makes any sense?

- We agree with this assessment and have now used the term “natural sampler” throughout the manuscript instead of “metaorganism”.

Line 58 – how do they know they are characterizing canopy-associated fungi and bacteria and not just an aggregation of all?

- The ESB samples just consist of leaves collected from the canopy (Tarricone et al. 2018a, Tarricone et al. 2018b, Klein et al. 2018), so we assume the recovered communities are canopy-associated. As mentioned above, we can already rule out that the microbial community is sourced from arthropods (as they represent at most 2% of the reads). We can, however, not rule out the possibility that some of the microbes are also sourced from the air.

Hypotheses – It feels like H2 contradicts H1?

- Yes, these hypotheses are contradicting each other, as they are alternative hypotheses. We have added this information in the text to make it clearer (lines 62-65).

Is it predicted that fungal diversity will be higher than bacterial diversity in the tree leaf sample data?

- Fungal diversity is not higher than bacterial diversity in our data, yet it is quite high. However, fungal diversity in trees is known to be extremely high, especially endophytic fungi are hyperdiverse, as for example shown in Zimmerman et al. 2012. The finding of high fungal diversity is hence not very surprising for us. However, one thing which is evident here, is that the mussel associated bacterial diversity is higher than that of the leaf and bladderwrack associated communities. One issue we were facing in our analysis of bacteria from plant samples was that the standard 16S bacterial primers also amplify chloroplast 16S. As these samples are mostly consisting of plant/algal tissue, the chloroplast DNA is massively overdominant over that of bacteria. We thus had to use primers, which prevent chloroplast amplification. These primers however, will most likely also show some bias against some bacterial groups (for example cyanobacteria). It is thus not too surprising that the recovered diversity is lower. We have added more information on this in the methods (lines 405-409).

Figure 1A is not introduced in the text?

- This has now been remedied (line 78).

Line 88 – how do these samples/analyses differ from the Krehenwinkel et al. eLife paper? Is this dataset completely novel, or are some data shared?

- As mentioned above, it’s an almost entirely novel dataset with some of the data shared. We have used a better suited extraction protocol and longer time series for this analysis. We have

added information on this in the methods (lines 353-358). Also, the Krehenwinkel et al. 2022 paper only looked at arthropods, while we here have expanded to bacteria and fungi.

Lines 94-95 – Is it likely that bacterial diversity is ca. 5K OTUs, whereas metazoan richness is 3.2K OTUs; it seems intuitive that either the bacteria are low and the metazoans are high. Do these data add up to the team – what do contemporary richness estimates look like from forest ecosystems. If this represents an unexpected skew, what explanations/tests could the team do/suggest?

- As mentioned above this may be associated with the primers we used for recovering tree and algae associated bacterial communities. It is evident that the bacterial diversity is considerably higher in mussels, probably due to primer bias in the plant and algal samples. We have added more information on this in the methods (lines 405-409). In addition, we used different marker genes for the different types of organisms (16S, COI), which accumulate mutations at different speeds. So comparing the OTU richness between datasets generated with different marker genes is not really informative as the different OTUs that are generated do not necessarily correspond to the same taxonomic level. This may also influence the comparatively high fungal diversity recovered.

Line 115 – Baltic, North, salinity data – no references?

- We have added this now (line 114).

Lines 142-143 – Is the model being used here novel to this paper, or has been published elsewhere? I could not see any of the team in the references cited? If the model is new, does it need to be benchmarked on sample/existing data to check the assumptions/parameters are working as expected?

- As explained above, the modelling approach we used is new and has been developed for the sake of this paper. It is however inspired from the previous theoretical work of two involved authors, BPL and HM (see Overcast et al. 2022).

We have now benchmarked our modelling approach using simulations (see above and in the Methods section, lines 581-597), which validate that everything is working as expected (see Extended Data Table 2).

All the scripts of this new modelling approach will be freely available on an online repository upon publication, and the method could be used in other systems by future studies.

Line 151 – strongly – compared to what?

- Richness of limnic prokaryotes strongly increased in comparison to all other (increasing) trends. We hope this gets clear from the context and Figure 4E (lines 148-150).

Line 155 – individual?

- Across individual sites. Sorry - we missed a word here. We now changed it to “across sites”. Thanks for recognizing (line 153).

Line 180 – changing env.conditions – which ones, based on what data?

- Thank you for your attention to the details. Since we did not include any data about changing environmental conditions, we cannot make any statement about this. We meant to express an assumption based on earlier studies and examples therein. We moved this part to the discussion, rephrased and referenced it accordingly (lines 261-269).

Line 203 – Danube, Rhine biogeography – no reference cited?

- Reference was added (line 286).

Line 227 – stability and functional redundancy – is there any evidence that the ecosystems are unstable?

- Thanks again for your attention to the details. Since ecosystem stability is not addressed or analyzed in this manuscript this part was deleted.

Line 391 – All these taxa is really vague – can you provide more information?

- “All these taxa” refers to the community associated with bladderwrack. See Figure 2 for the taxonomic composition and Supplemental Table 3 for single species.

Line 415 – check journal style wrt numbers and in text written citations.

- Thank you. We have now ensured that the references meet the criteria of the NEE author guidelines.

Line 416 – under(neath?) clean benches – do they mean on clean benches?

- Yes, on clean benches. This was corrected (line 392).

Line 418 – 200mg – does reference 9 only refer to the leaf samples; if so, this needs to be clear.

- For all samples 200mg of tissue/plant material was used for isolation. The reference 9 only refers to the required material to achieve an adequate representation of the sample-associated diversity in trees. However, this amount was also found to be suitable in other recent work for mussels and bladder wrack (Weber et al. 2023). Also, as all sample types are ground in the same way (Rüdel et al. 2008), we assume that the results of the weight series presented in Krehenwinkel et al. (2022) are also valid for all other species. We added this information to the methods (line 393-396).

Line 421 – barcode – metabarcode?

- We changed it to metabarcode marker now (line 399).

Line 462 – Wed excluded?

- Misspelling: We excluded all taxa except Bacteria, Algae/Protozoa, Metazoa or Fungi from the respective datasets (line 444).

Line 466 – checked for contamination using negative controls – how? What data (or none?) were in the negatives? Line 478 – similar question. Where did the 0.01% threshold come from – the negatives, or random number?

- The negative controls were used to check for contamination by removing OTUs present in the negative controls from the dataset. This was the case, for example, with some typical laboratory contaminants or dermal bacteria that were detected in the negative controls. However, in the samples, these taxa were only found in very low read numbers or not at all and thus excluded from the analysis. Conversely, some highly abundant OTUs were also found in the negative controls in very low read numbers. This is likely caused by carry-over during laboratory procedure or sequencing, which is why we kept these sequences in the datasets. We added more details to the removal of contamination in the methods (lines 455-461).
- We used a 0.01 % cutoff for individual samples to deal with these carryover contaminants. Since there is no standardization on such thresholds, 0.01 % was chosen randomly. However, we tried different cutoffs beforehand and it did not affect the results. Furthermore, the aforementioned OTUs detected in samples and negative controls comprise an averaged 0.016 % of the reads per sample across all datasets/communities. We are thus confident that the chosen 0.01 % cutoff well excludes carryover.

Lines 481-487 – how many samples were involved in these comparisons?

- 537 samples. We added the information in the methods (line 471).

Line 563 – year 1500 – why?

- Our model is an equilibrium model of community assembly based on the equilibrium theory of island biogeography (MacArthur & Wilson 1963). So during the simulation procedures, we need to simulate the on-going ecological processes for a sufficient duration to reach an equilibrium. Preliminary analyses (not shown) have revealed that simulating the processes for more than 5 centuries (i.e. starting in the year 1500) is sufficient to reach an equilibrium in most cases. We have now clarified this in the Methods section (line 554-558).

References

Blowes, S. A. *et al.* Synthesis reveals approximately balanced biotic differentiation and homogenization. *Science Advances* **10**, eadj9395; 10.1126/sciadv.adj9395 (2024).

Klein, R., Tarricone, K., Teubner, D. & Paulus, M. Guideline for Sampling and Sample Processing Norway Spruce (*Picea abies*)/ Scots Pine (*Pinus sylvestris*). *Umweltbundesamt* (2018).

Krechenwinkel, H. *et al.* Environmental DNA from archived leaves reveals widespread temporal turnover and biotic homogenization in forest arthropod communities. *eLife* **11**, e78521; 10.7554/eLife.78521 (2022).

Krechenwinkel, H., Weber, S., Künzel, S. & Kennedy, S. R. The bug in a teacup—monitoring arthropod–plant associations with environmental DNA from dried plant material. *Biology Letters* **18**, 20220091; 10.1098/rsbl.2022.0091 (2022).

MacArthur, R. H. & Wilson, E. O. An Equilibrium Theory of Insular Zoogeography, *Evolution* **17**, 373–387; 10.2307/2407089 (1963).

Overcast, I. *et al.* Towards a genetic theory of island biogeography: Inferring processes from multidimensional community-scale data. *Global Ecol Biogeogr* **32**, 4–23; 10.1111/geb.13604 (2023).

Rüdel, H., Uhlig, S. & Weingärtner, M. https://www.umweltprobenbank.de/upb_static/fck/download/IME_SOP_Probenvorbereitung_Dez2008_V200.pdf. *Umweltbundesamt* (2008).

Tarricone, K., Klein, R., Paulus, M. & Teubner, D. Guideline for Sampling and Sample Processing Lombardy Poplar (*Populus nigra* ‘Italica’). *Umweltbundesamt* (2018).

Tarricone, K., Klein, R., Paulus, M. & Teubner, D. Guideline for Sampling and Sample Processing Red beech (*Fagus sylvatica*). *Umweltbundesamt* (2018).

Zhao, J., Becker, P. R. & Meng, X.-Z. 2013 International Conference on Environmental Specimen Banks: Securing a Strategy to Monitor Emerging Pollutants in the Regional and Global Environment. *Environmental Science and Pollution Research* **22**, 1555–1558; 10.1007/s11356-014-3715-9 (2015).

Zimmerman, N. B. & Vitousek, P. M. Fungal endophyte communities reflect environmental structuring across a Hawaiian landscape. *Proceedings of the National Academy of Sciences of the United States of America* **109**, 13022–13027; 10.1073/pnas.1209872109 (2012)